# Carboxylic acids from limonene oxidation by ozone and OH radicals: Insights into mechanisms derived using a FIGAERO-CIMS

Julia Hammes[1], Anna Lutz[1], Thomas Mentel[1,2], Cameron Faxon[1], Mattias Hallquist[1*]

[1] Department of Chemistry and Molecular Biology, University of Gothenburg, Gothenburg, Sweden
[2] Institute of Energy and Climate Research, IEK-8: Troposphere, Forschungszentrum Jülich GmbH, Jülich, Germany
*Correspondence to*: Mattias Hallquist (hallq@chemgu.se)

**Abstract.** This work presents the results from a flow reactor study on the formation of carboxylic acids from limonene oxidation in the presence of ozone under NOx free conditions in the dark. A High Resolution Time Of Flight acetate Chemical Ionisation Mass Spectrometer (HR – TOF – CIMS) was used in combination with the Filter Inlet for Gases and AEROsols (FIGAERO) to measure the carboxylic acids in the gas and particle phases. The results revealed that limonene oxidation produced large amounts of carboxylic acids which are important contributors to secondary organic aerosol (SOA) formation. The highest 10 acids contributed 56–91% to the total gas-phase signal and the dominant gas-phase species in most experiments were $C_8H_{12}O_4$, $C_9H_{14}O_4$, $C_7H_{10}O_4$ and $C_{10}H_{16}O_3$. The particle-phase composition was generally more complex than the gas-phase composition and the highest 10 acids contributed 47–92% to the total signal. The dominant species in the particle phase were $C_8H_{12}O_5$, $C_9H_{14}O_5$, $C_9H_{12}O_5$ and $C_{10}H_{16}O_4$. The measured concentration of dimers bearing at least one carboxylic acid function in the particle phase was very low, indicating that acidic dimers play a minor role in SOA formation via ozone/OH oxidation of limonene. Based on the various experimental conditions, the acidic composition for all experiments were modelled using the descriptions from the Master Chemical Mechanisms (MCM). The experiment and model providing yield of large (c7-c10) carboxylic acid in the order of 10% (2-23 and 10-15%, respectively). Significant concentrations of 11 acids, from a total of 16 acids, included in MCM were measured with the CIMS. However, the model predictions were, in some cases, inconsistent with the measurement results, especially in the case of the OH dependence. Reaction mechanisms are suggested to fill-in the knowledge gaps. Using the additional mechanisms proposed in this work nearly 75% of the observed gas-phase signal in our lowest concentration experiment (8.4 ppb converted, ca 23% acid yield) done at humid conditions can be understood.

# 1 Introduction

Atmospheric aerosol particles have an impact on climate and human health and the respective effects depend on particle properties determined by the size and its chemical composition. Among the many constituents of atmospheric aerosol particles, organic aerosol particles are the least understood (Glasius and Goldstein, 2016).
Secondary organic aerosol (SOA) is the major component of organic aerosols. Identifying the chemical pathways of condensable products is essential for predicting SOA formation (Hallquist et al., 2009;Ziemann and Atkinson, 2012;Ehn et al., 2014;Shrivastava et al., 2017;McFiggans et al., 2019). However, this identification is inherently difficult as such products often reside in both the gas and particulate phases and continuous partitioning occurs between these two phases. Low vapour pressure products from radical- (i.e. OH) initiated oxidation or ozonolysis of volatile organic compounds (VOCs), such as monoterpenes ($C_{10}H_{16}$), contribute significantly to atmospheric aerosol particle formation and growth (Hallquist et al., 2009). Limonene, the main constituent of the essential oil from citrus plants, is a widely used chemical in personal care and household-related consumer products (owing to its pleasant smell) and elevated indoor concentrations can be expected (Brown et al., 1994;Langer et al., 2008) with subsequent SOA formation (Youssefi and Waring, 2014). The total global forest emission of limonene has been estimated to 11.4 Tg year$^{-1}$, placing it on the top four among monoterpenes (Guenther et al., 2012). A high aerosol yield and the two chemically different double bonds, an endocyclic and an exocyclic double bond makes limonene ozonolysis of specific interest (Koch et al., 2000;Saathoff et al., 2009;Chen and Hopke, 2010;Gong et al., 2018). The initial reaction will occur predominantly at the endocyclic double bond. However, the first generation products may be unsaturated and exhibit high reactivity for further oxidation. The oxidation of limonene will eventually lead to the formation of SOA in both the atmosphere and indoor environments. The oxidation of monoterpenes and, specifically, limonene has been previously reported (Leungsakul et al., 2005a;Walser et al., 2008;Leungsakul et al., 2005b;Maksymiuk et al., 2009) and reaction mechanisms that describe first- and second-generation oxidation products have been proposed (Carslaw, 2013;Chen and Griffin, 2005). Due to their low vapour pressure, carboxylic acids, a major class of limonene-oxidation products can play an important role in SOA formation (Salo et al., 2010). The relative contribution of carboxylic acids from limonene oxidation to SOA formation has been assessed via a model (Pathak et al., 2012). According to that study, limonene-ozonolysis produces significant amounts of carboxylic acids and the distribution of these acids is affected by the OH and ozone concentrations.

During ozonolysis, limonene is attacked by ozone and forms an unstable and energy-rich primary ozonide (POZ), see Figure 1. The POZ will undergo decomposition where the oxygen atoms contribute to the formation of a carbonyl and a carbonyl oxide group, the so-called excited Criegee intermediate (CI*). The 10 carbon skeleton is

retained during this process, if an endocyclic double bond is attacked. The CI* has a planar structure and the orientation of the outer oxygen will determine its chemical fate. The dominant reaction pathway (86% (Atkinson et al., 1992)) for limonene syn – CI* is the vinyl hydroperoxide channel (VHP) which generates an alkyl radical under loss of an OH radical. This pathway provides a source for night-time OH in the atmosphere. The VHP
requires an alkyl group in the syn position and is, hence, inaccessible to anti – CI*. The dominant fate of the anti – CI* is decomposition via the ester or the "hot acid" channel where an energy-rich ester or acid formed will undergo decomposition thereby resulting in various products. Two possible products, i.e. OH and an acyl radical $(RC(O)•)$ (Vereecken and Francisco, 2012), which can react with $O_2$ and subsequently $HO_2$ to form a carboxylic acid and ozone. Furthermore, the CI* can, to some extent, become collisionally stabilised (sCI) and exocyclic CI* are
stabilised more efficiently than endocyclic CI*. The formed sCI will undergo further reactions and the reaction sCI + water will produce a carbonyl, an alkyl or an alkoxy radical. If the sCI contains an α-hydrogen, a carboxylic acid can be produced directly from the water reaction. Although the sCI + water reaction is likely the most dominant in the atmosphere, sCI may also react with carboxylic acids forming stable adducts which have been identified as dimer esters (Kristensen et al., 2016). The decomposition of CI* can lead to the formation of alkyl radicals.  They
rapidly react with oxygen to form alkylperoxy radicals $(RO_2)$ which are an important intermediate in the gas phase oxidation of organic compounds.

The atmospheric fate of $RO_2$ radicals in the absence of $NO_x$ includes a self-reaction (reaction (1) – (3)), isomerisation via an internal H-shift (reaction (4)), and a reaction with $HO_2$ (reaction (5)) (Ehn et al., 2014). If $RO_2$ is an acylperoxy radical, a carboxylic acid can be formed.

$RO_2 • + RO_2 • \rightarrow RO • + O_2$       (1)

      $\rightarrow ROH + RCHO + O_2$     (2)

      $\rightarrow ROOR + O_2$     (3)

$RO_2 •$ [internal H − shift] $\rightarrow • ROOH$     (4)

$RC(O)O_2 • + HO_2 • \rightarrow RC(O)OH + O_3$     (5)

These reactions lead to further functionalisation, e.g. formation of acids, alcohols, carbonyls or peroxides and may in addition produce alkoxy radicals. Subsequently, alkoxy radicals can be converted by oxygen to a carbonyl, if an α-hydrogen is present. Alkoxy radicals that lack this hydrogen will undergo isomerisation or decomposition via β-scission.

During ozonolysis experiments, OH radicals are produced and react with the precursor as well as the reaction
products. This process occurs in the laboratory as well as in the actual atmosphere and increases the complexity of

the degradation mechanisms. In the laboratory, one can scavenge the produced OH radicals by adding a compound, e.g. 2-butanol, that reacts rapidly with OH, thereby reducing OH. The OH scavenger reduces the OH concentration but leads to an increase in the $HO_2$ and $RO_2$ concentration. This yields changes in the distribution of radicals and subsequently the radical-dependent chemistry (Keywood et al., 2004;Jonsson et al., 2008b). For example, the reaction of 2-butanol with OH produces $HO_2$ radicals with a yield of 64% (MCM v 3.3.1) thereby increasing the $HO_2/RO_2$ ratio. In laboratory experiments, these features can be employed in investigating the importance of various radicals/pathways for product distribution and subsequent SOA formation.

The Gothenburg Flow Reactor for Oxidation Studies at Low Temperatures (G-FROST) has been used in previous studies (Jonsson et al., 2006;Faxon et al., 2018;Jonsson et al., 2008a;Jonsson et al., 2008b;Kristensen et al., 2014) to investigate the dependence of aerosol properties on different parameters (e.g. humidity and radical conditions). The G-FROST setup has now been extended with a High-Resolution Time-of-Flight Chemical Ionization Mass Spectrometer (HR-ToF-CIMS) that will provide insight into the chemical composition of the gas and particle phase through connection to the Filter Inlet for Gases and AEROsols (FIGAERO). These new techniques allow for sensitive simultaneous detection in the gas and particle phases. Herein, an ionisation using acetate allows investigation of carboxylic acid formation. In the following, we analyse the carboxylic acid product spectrum of limonene. The goal is to detect major pathways and to compare the results with a model using the existing master chemical mechanism (MCM) that was primarily developed for gas-phase chemistry related to the impact on tropospheric ozone formation, but now frequently are used as a link to particle formation. This work (i) considers ozonolysis under dark condition and NOx free conditions (for various limonene concentrations) the effect of humidity, OH scavenging and ozone level on carboxylic acid formation, and (ii) provides an outlook and suggestions for mechanistic gaps with the aim of eventually describing major acidic products found in the gas and particle phases under realistic atmospheric conditions, i.e. ozonolysis is performed in the absence of an OH scavenger under low concentration and humidity conditions.

## 2 Materials and methods

Oxidation studies of limonene in the presence of ozone have been performed under a variety of experimental conditions (Table 1). The experimental matrix was chosen to understand ozonolyis of limonene under dark NOx free conditions. The laminar flow reactor approach is well suited for investigating changes in experimental condition, e.g. dry vs humid, high vs low concentration. Due to short residence time the absolute concentration is

higher than ambient even if the amount limonene converted (few pbb and higher) is similar to larger simulation chamber studies. The conversion of any $RO_2$ radicals is biased towards self-reaction; of importance in VOC dominated rural forest conditions and in the indoor environment. The G-FROST system employed has been described in detail elsewhere (Jonsson et al., 2008b;Jonsson et al., 2008a) and will only be presented briefly here.

G-FROST consists of a laminar-flow reactor (vertical Pyrex glass cylinder, length 191 cm, inner diameter 10 cm, with a halocarbon wax coating) in a temperature-controlled housing (see Figure S1). The total inflow into the system was 1.6 LPM and the sample outflow was 0.94 LPM, yielding an average residence time of 240 s. The aerosol was sampled with a funnel system from the centre part of the laminar flow, to minimise wall effects. Limonene (Alfa Aesar, (R)-(+)-Limonene, 97%) was added by passing synthetic air (Laboratory Zero Air

Generator, N-GC-6000, Linde Gas) through a characteristic diffusion source. Limonene was then pre-mixed with a dry or humidified bulk flow, with or without 2-butanol (Merck, p.a. >99%) as an OH scavenger. During each experiment, limonene concentrations were increased stepwise (15, 40, 150 ppb), while the temperature inside G-FROST was kept constant at 20°C for either dry (relative humidity (RH) <2%) or 40% RH conditions. Ozone (400, 1000, 5000 ppb) was generated by passing oxygen gas through a set of Pen-Ray® mercury lamps (UVP, λ 254 nm)

and added through a separate 6-mm Teflon line to G-FROST. The ozone level was kept constant during each experimental run.

A summary of experimental conditions is provided in Table 1. The product distribution in the gas and particle phases was analysed with an acetate HR-ToF-CIMS (Aerodyne) (Bertram et al., 2011) coupled to the FIGAERO inlet (Lopez-Hilfiker et al., 2014). The reagent ion acetate is especially susceptible to acidic organic compounds

such as carboxylic acids (Bertram et al., 2011). One may note that also proxy acids has high sensitivities (Lopez-Hilfiker et al., 2015) and that the acetate ionisation has previously been used to detect nitro phenols (Mohr et al., 2013;Le Breton et al., 2019) and organic sulphates (Le Breton et al., 2019). However, here we assume carboxylic acids and peroxy acids to be the primarily compounds being observed with current set-up the CIMS. The used sensitivity for larger carboxylic acid was 5.5 x $10^{-3}$ Hz $ppt^{-1}$ (Le Breton et al., 2019). This sensitivity was used to

estimate molar yields; even if one should be precautious to provide absolute yields from this type of studies it provides indications on the product contribution. The sample flow from G-FROST was diluted with ultra-high purity (UHP) nitrogen gas and pumped at 2 × 4 LPM by two diaphragm pumps (KNF, N816.3KN.18) through the FIGAERO inlet. The dilution was necessary for analytical reason and might evaporate some of the most volatile compounds from the condensed phase. However, the time between dilution and analysis was short (< 1s), exactly

the same for all conditions and a possible effect would somewhat mimic atmospheric conditions. Perfluoroheptanoic acid (Sigma Aldrich, 99%) was used as the internal mass calibration standard. The gas-phase

composition was determined via 60 min measurements and particles were collected simultaneously on a 1 µm 24 mm Zefluor® PTFE filter (Pall Corp.). During desorption, the temperature was increased from 25°C to 200°C in 50 min (3.5°C min$^{-1}$) and kept constant at 200°C for 10 min. Subsequently, UHP nitrogen gas was bubbled (flow rate: 0.02 LPM) through acetic anhydride (Sigma Aldrich, puriss p.a. ≥99%) and diluted with a bulk flow of UHP nitrogen to 2.2 LPM. This flow was reduced to 2 LPM using a critical orifice (O'Keefe Controls Co) and passed through a commercial $^{210}$Po alpha emitter (NDR, P-2021) to produce acetate reagent ions. A Scanning Mobility Particle Sizer (SMPS; CPC 3775 and DMA 3081, TSI Inc.) was used to measure the particle size distribution. The mass of the produced aerosol was determined, assuming a particle density of 1.4 g cm$^{-3}$ (Hallquist et al., 2009). The CIMS data were analysed using the Tofware package (Tofwerk/Aerodyne) for IGOR Pro (WaveMetrics). The data were acquired at 1 Hz and pre-averaged to 0.0167 Hz (1 min) for further analysis. To account for thermal decomposition (double or triple peaks in desorption profile), the average (four desorption cycles per reaction condition) FIGAERO desorption profiles (thermograms) were analysed in Python 3.6.0 using the NumPy (v 1.11.3), SciPy library (v 0.18.1) and pandas (v 0.19.2) packages. The exponentially modified Gaussian function (Foley and Dorsey, 1984) was used as a peak shape function for peak fitting of the thermograms (Fig. S2). The area of the fitted peaks was calculated by integrating along the given axis using the composite trapezoidal rule. A spearman correlation analysis was done based on of major products, experimental conditions and calculated radical concentrations. Compared to standard correlation the spearman correlation is more robust to outliers and independent of any assumptions about the distribution of the data. It was therefore preferred to assess the degree of association between each dominant acid and experimental parameters. The evaluation using spearman correlation is similar to other correlation methods giving 0, -1 and 1 for no correlation, perfect negative and positive correlation, respectively. All experiments have been modelled utilizing an open access mechanism (MCM v3.3.1) (see Table 1 for the initial conditions). The initial concentration of 2-butanol was set to $3 \times 10^5$ µg m$^{-3}$ in the case of OH scavenger experiments (Pathak et al., 2012). Based on the calculations, the amount of reacted limonene was derived. The OH, HO$_2$ and RO$_2$ levels enabling calculation of the corresponding values integrated over a reaction time of 240 s were used in the spearman correlation analysis.

## 3 Results and discussion

A total of 33 different experiments have been performed under various reaction conditions (Table 1). In the following, we will characterise the distribution of gas- and particle-phase organic acid. Figure 2 shows an example of a mass spectrum from one of the experiments. This experiment was done at medium ozone and high limonene

concentration with an estimated 8 % molar yield of large carboxylic acids. Still over 100 different molecular formulas for acids have been identified, far exceeding the number of acids reported in previous studies (Leungsakul et al., 2005a;Jaoui et al., 2005;Jaoui et al., 2006;Rossignol et al., 2012;Rossignol et al., 2013;Walser et al., 2008;Marianne Glasius et al., 2000). Here we will focus on the analysis of acids with carbon numbers ranging from seven to ten (and the dimers formed from these acids). Typically, these acids stand for 2-23% of reacted limonene on a molar basis assuming an average reaction time and CIMS sensitivity (Table 1). The distribution between gas and particle phase varied between compounds where the average particle fraction was between 5 and 80% depending on experimental conditions. The contribution of each acid to the total signal is calculated and the highest 10 ion signals are selected from each experiment. This gave a total of 32 different molecular compositions, representing the greatest fraction (47%–91%) of the total signal

The fraction corresponding to the sum of 10 highest ions to the total signal can reveal the diversity of the product distribution for each condition. A low coverage indicates an experiment where several compounds with the same intensities are generated. Figure 3 shows for the comparable data to Fig. 2 (medium ozone, high limonene) the fraction of the 10 most prevalent ions. This was the most complete sub-step (dry-humid, with and without scavenger) of the total matrix and the illustrated pairwise features (e.g. dry vs humid) that was common also among other concentrations (e.g. for other condition we had missing particle phase data, see table 1). For these four selected experiments the estimated total yield of larger carboxylic acids (c7-c10) were very similar 6% (dry, scavenger), 8% (humid, scavenger), 6% (dry, no scavenger) and 7 % (humid, no scavenger).

In general, the particle-phase composition is more diverse, less dominated by the 10 top compounds, than the gas-phase composition (Figure 3). The presence of water in the system also increases the diversity of the product distribution in both the gas and particle phases, i.e. a small effect in Fig. 3 but consistent for all pairwise dry/humid experiments shown in Table 1. Compared with lower ozone concentrations, higher concentrations generally gave larger product diversity, owing to greater possibility for exocyclic double-bond oxidation or unsaturated-acid oxidation that yields a wider variety of products. The OH reaction pathways are suppressed in experiments with an OH scavenger and the oxidation can then occur only via ozonolysis. This apparently reduces the oxidation product diversity of the particle phase, consistent with the findings of Watne et al. (Watne et al., 2017). In that study, the volatility of limonene SOA produced via ozonolysis only was found to be more homogeneous than that of limonene produced via other/additional processes.

Figure 4 shows the total molar yield of the acids identified in this study. One may note that the absolute yield presented here has significant uncertainties while the relative important of an acid can guide us in our mechanistic interpretation. Generally, the most important acids (averaged over all experiments) are $C_7H_{10}O_3$, $C_7H_{10}O_4$, $C_8H_{12}O_4$,

$C_8H_{12}O_5$, $C_9H_{14}O_4$, $C_9H_{14}O_5$, $C_{10}H_{16}O_3$ and $C_{10}H_{16}O_4$ with yields at or below 1% which is in line with the yeidl of major acids in the study of Glasius et al. (2000). The most important acid in the present study are compared with an overview (Table S1) of previously reported carboxylic acids ($C_7$–$C_{10}$) resulting from limonene ozonolysis. Table S1 also illustrate the proposed structures of these acids based on the current literature. Ten of the previously reported acid formulas are found in this study while three acids, $C_7H_{10}O_6$, $C_8H_{12}O_6$ and $C_8H_{14}O_4$, lie outside the ten highest corresponding ions identified in any of our 33 experiments. Leungsakul et al. (Leungsakul et al., 2005a) and Walser et al. (Walser et al., 2008) reported that $C_9H_{14}O_4$ and $C_{10}H_{16}O_3$ were the most and second-most dominant particle-phase compounds, respectively. However, in our study, the more oxidised (compared with $C_9H_{14}O_4$ and $C_{10}H_{16}O_3$) compounds $C_9H_{14}O_5$ and $C_{10}H_{16}O_4$ are the dominant in the particle-phase. From Fig. 4 it is evident that the composition of acids is very complex and many compounds contributes to the total signal. Also among the most prominent acids there is no acid that clearly dominate in any of the experiments. We should also remember that in addition to the acids there are many more product categories contributing to the product distribution, e.g. recently, Gong et al. (2018) did a similar study focusing on peroxides and carbonyls. Utilising the complexity to further understand the mechanism requires some strategy. Here we decide to start with the 10 most prominent ions for each experiment creating a correlation matrix of 32 different molecular compositions, where each composition might include several isomers. The intensities measured for each compounds are presented in the supplemental, Table S2 and S3. In Fig. S3-S4, we show the corresponding correlations, using Spearman ranking, for each of the 32 different molecular compositions representing the majority of the ion signals divided in with and without OH scavenger (Fig S3) and for humid and dry conditions (Fig S4). The results for the eight most important acid formulas, i.e. $C_7H_{10}O_3$, $C_7H_{10}O_4$, $C_8H_{12}O_4$, $C_8H_{12}O_5$, $C_9H_{14}O_4$, $C_9H_{14}O_5$, $C_{10}H_{16}O_3$ and $C_{10}H_{16}O_4$, are presented and discussed. These are all oxidation products with mass ranging from 130 $m/z$ to 250 $m/z$ and are identified as carboxylic acids with carbon numbers ≤10. Based on other studies (Kristensen et al., 2014;Mohr et al., 2017;Kristensen et al., 2012;Kristensen et al., 2016;Witkowski and Gierczak, 2014), dimer formation is expected. These dimers are expected to contribute significantly to the particle phase. For the particle-phase data, compounds with mass above 300 $m/z$ are detected and are classified as dimer species if they have a carbon numbers >10. These compounds occur only in the particle phase. However, the relative signals are significantly lower than those reported for dimer formation in a study on limonene with nitrate radicals (Faxon et al., 2018) or the ozonolysis of other terpenes such as α-pinene (Kristensen et al., 2016). In the present study the identified products must be acids, since we apply CI using the acetate ion. We conclude that, although dimer formation may occur (in general), no

important acidic dimers are formed in the system. Consequently, we will focus on the formation of the monomer acids.

**Water effect.** Generally, most of the 32 top ions have higher signals in humidity experiments than in other environments, Table S2 and S3 and correlation matrix in Fig. S4. The opposite is true for the 400 ppb ozonolysis-only (OH-scavenged) cases (gas and particle phases). For experiments with OH scavenger, the importance of water is evidenced by the prominent formation of gas-phase $C_{10}H_{16}O_3$, $C_9H_{16}O_3$, $C_9H_{14}O_3$ and $C_8H_{14}O_3$ (4). The water dependence of these acids is less pronounced in the mixed oxidation cases (except for $C_8H_{14}O_3$), than in other cases, but water seems to be favourable for the formation of other acids, such as $C_8H_{10}O_{4-5}$ and $C_7H_{10}O_{2-3}$. In general, water enhances the formation of the particle-phase acids. This concurs with the findings of Jonsson et al. (Jonsson et al., 2006) who reported an increase in the SOA number and mass under humid conditions. The authors attributed (i) this result to an increase in the number of low-volatility products with increasing water concentration during the ozonolysis of limonene, and (ii) the water effect on SOA formation to $C_{10}H_{16}O_3$ formation. For humidity experiments considered in the present study, we observe a considerable increase and a slight increase in the formation of gas-phase $C_{10}H_{16}O_3$ and particle-phase $C_{10}H_{16}O_3$, respectively. Assuming that the humidity effect on $C_{10}H_{16}O_3$ production is responsible for the SOA dependence on humidity, subsequent transformation of condensed material is required as the particulate phase is deficient in $C_{10}H_{16}O_3$.

**Radical effect.** Consistent with previously reported results on the SOA mass (Jonsson et al., 2008b;Pathak et al., 2012), the intensities of most acids in the low- and medium-ozone cases are higher for experiments employing mixed oxidation than for experiments employing an OH scavenger. For low-ozone and low-VOC experiments, the scavenger-provided SOA mass decreases with 2-butanol addition, as previously reported (Jonsson et al., 2008a), although the effect observed here is weaker than the effect reported in that work. However, for relatively high concentrations of limonene, the opposite effect is observed, i.e. the SOA mass increases with the use of a scavenger. Notably, this effect occurs independently of the acid-intensity behaviour, and may have resulted from the fact that (i) the SOAs associated with mixed oxidation are quite volatile and (ii) increased oxidation in the presence of OH, rather than converting semi volatiles to low/extremely low volatiles, converts volatiles/intermediate volatiles to semi volatile species, as suggested by Pathak et al. (2012); these are then lost during the dilution process. Another possibility is that changes in the chemistry affect nucleation, as indicated by a size-distribution shift to smaller sizes which (compared with larger sizes) are more susceptible to evaporative losses in the dilution step. Separation of these effects during the experiments is impossible and, hence, the SOA formation potential associated with mixed oxidation may have been underestimated in this study. Owing to the sufficiently low ozone levels employed in the

low and medium experiments, OH has an influence on the reaction pathways. At the highest ozone level, however, the intensities of acids associated with mixed oxidation are lower than those resulting from the use of an OH scavenger. To investigate the effect of radical chemistry on the reaction pathways leading to the observed carboxylic acids, the OH, $HO_2$ and $RO_2$ concentrations are calculated and integrated using the model for each experiment (Table 1). Regarding correlation, Fig. S3, a comparison of the mixed oxidation cases reveals that the formation of most gas-phase acids (e.g. $C_{10}H_{16}O_3$, $C_9H_{14}O_4$ and $C_7H_{10}O_3$) decreases with increasing amount of OH radicals in the system. The $HO_2/RO_2$ ratio has only a small influence on the mixed oxidation. However, when an OH scavenger is used, the amount of gas-phase products ($C_{10}H_{16}O_3$, $C_{10}H_{16}O_4$, $C_9H_{14}O_4$ and $C_8H_{14}O_3$) decreases considerably with increasing $HO_2/RO_2$. The general influence of OH on acid formation is most pronounced for experiments performed under dry conditions. Under these conditions, OH and $HO_2/RO_2$ have a significant effect on the formation of $C_{10}H_{16}O_3$, $C_9H_{14}O_3$ and $C_7H_{10}O_4$. For example, $C_{10}H_{16}O_3$ and $C_9H_{14}O_3$ formation increases with increasing OH and decreasing $HO_2/RO_2$. The opposite is true for $C_7H_{10}O_4$ formation which decreases with increasing OH and decreasing $HO_2/RO_2$.

**Effect of excess ozone**. Experiments with high ozone levels are performed to assess the effect of excess ozone on acidic oxidation products. The aim is to oxidise, with ozone, the remaining double bond of the produced unsaturated carboxylic acids. The results show that ozone has a distinct negative impact on $C_7H_{10}O_{2-3}$ in the pure ozonolysis cases (see correlation matrix in Fig. S3) and, hence, we conclude that those compounds are unsaturated.

Furthermore, the levels of gas-phase $C_{10}H_{16}O_3$, $C_9H_{16}O_3$, $C_9H_{14}O_{3-4}$ and $C_8H_{14}O_3$ are positively correlated with ozone in the absence of OH. For $C_{10}H_{16}O_3$, this is surprising as this compound is assumed to be limononic acid, an unsaturated compound. This positive correlation may have resulted from the fact that the production of $C_{10}H_{16}O_3$ dominates over the removal (via ozonolysis) of the remaining double bond. The correlation with ozone is negative for most acids in the presence of OH and is most pronounced for gas-phase $C_{10}H_{16}O_3$, $C_9H_{14}O_3$ and $C_8H_{12}O_3$. The

negative ozone correlation observed for mixed oxidation cases considering $C_{10}H_{16}O_3$ and $C_9H_{14}O_3$ concurs with the modelling results of a previous study that assessed the influence of ozone on limonene oxidation (Pathak et al., 2012). A positive (albeit slightly) correlation with ozone is observed only for particle-phase $C_8H_{14}O_3$. The acid-ozone correlation obtained for humid conditions differs significantly from that obtained for dry conditions. The negative acid-ozone correlation is quite pronounced in the dry experiment cases and becomes increasingly negative

(in general) for acids with relatively low carbon numbers, a trend unique to these experiments. The level of $C_{10}H_{16}O_3$ (especially the particle-phase) is positively correlated with ozone levels in the dry experiments. Generally, the amount of gas-phase acids has a stronger positive correlation with the limonene consumption ($\Delta$limonene) under humid conditions compared to dry experiments. In the dry experiments, $C_{10}H_{16}O_5$ and $C_9H_{14}O_4$ are the only acids with a strong positive correlation to $\Delta$ limonene. Compared with the occurrence of gas-phase acids, the

occurrence of particle-phase acids is (in general) more strongly correlated with $\Delta$ limonene.

**Model results and comparison with experiments.** Model calculations using the scheme presented by the master chemical mechanism (MCM, http://mcm.leeds.ac.uk/MCM/, v3.3.1) have been performed for all 33 experimental conditions, in order to calculate $\Delta$ limonene and radical concentrations as well as product distributions, based on the experimental conditions. The model did only consider gas-phase scheme of MCM. Most of the previously

reported carboxylic acid molecular formulas (see Table S1) are included in the MCM which was originally developed to provide accurate, robust and current information regarding the role of specific organic compounds in ground-level ozone formation, in relation to air-quality policy development in Europe. Over the years, MCM has been employed for models in studies linked to SOA formation (Jenkin, 2004), although this mechanism is still under development to capturing descriptions on the fraction of low-volatility and often very oxygenated organic

compounds (Barley et al., 2011).

Generally, the model provides small variation in the total molar yield for the large carboxylic acids (c7-c10) of 10-15% while the experiments shows larger variability (2-23% molar yield) plausible due to the complication of aerosol formation not covered by the model. A key for understanding the chemical mechanism leading to various products is the radical distribution. The experiments set-up where product distribution was measured at the end of the flow reactor restrict dynamic information. However, the variation of radical distribution between experiments is illustrated in Table 1. Here the values of radicals are given as the integral concentration over the reaction time (unit pbb × s). Furthermore, the integral $HO_2/RO_2$ ratio is presented together with a rate normalised ratio of these reaction, i.e. , the $HO_2 + RO_2$ reaction is rapid and the typical rate constant is one order of magnitude larger than that of the $RO_2 + RO_2$ self-reaction (Orlando and Tyndall, 2012).

Regarding oxidant/radical variation the modelled OH levels decrease with the initial limonene concentration, except for the highest ozone cases. High-ozone experiments yield the highest OH dose. The model results show that the $HO_2/RO_2$ ratio in experiments employing the OH scavenger 2-butanol is approximately one order of magnitude higher than that of the mixed oxidant experiments. This higher ratio results from the $HO_2$ radicals generated by the reaction of 2-butanol with OH and will provide more influence of the $HO_2 + RO_2$ reaction in the experiment with scavenger. However, the $RO_2$ self-reaction are still the major pathway also in these experiment, twice the normalised rate of the $HO_2$ reaction. One may note that the $RO_2$ reaction rates are very much structural dependent and might be faster or slower that the assumed rates, see Jenkin et al. (2019) for a recent review on $RO_2$ chemistry.

In the MCMv3.3.1, 25 closed-shell carboxylic acids with 16 different chemical formulas are included for limonene. We identify 11 of the 16 acids (Table S4 and S5; all MCM species used in the model are presented in Table S6). $C_9H_{14}O_3$ and $C_9H_{14}O_4$ are the most dominant and second-most dominant acids in all 33 modelled experiments. $C_{10}H_{16}O_3$ (LIMONONIC), formed by the reaction of sCI + water, is the only acid that exhibits an overall positive water dependence. The model calculations predict that water should also have a positive influence on KLIMONONIC and CO25C6CO2H. However, this influence is undetectable in our experiments, owing to the extremely low concentrations of these compounds and the stronger influence exerted on other compounds with the same molecular mass.

The model predicts that, compared with the presence of water, the presence of OH radicals has a greater influence on the product distribution. Most individual species from the MCM exert a strong positive OH-chemistry effect in the model, except for LIMONONIC ($C_{10}H_{16}O_3$), C823CO3H ($C_9H_{14}O_5$), C823OOH ($C_8H_{14}O_4$) and C825OOH ($C_8H_{12}O_5$). In all cases, the concentrations estimated with the model of the last three compounds are highest when the OH chemistry is "turned off" (2-butanol added in model). C82CO2H ($C_9H_{14}O_3$) is produced to a lesser extent

under humid and high ozone and for the highest OH conditions. It was produced to a higher extent under medium and low ozone and for the medium and lowest OH conditions. In the presence of OH chemistry, the LIMONONIC concentration is lower under humid conditions than under dry conditions. The presence of OH is essential for the formation of numerous compounds and yields significant concentration only in the absence of 2-butanol, i.e. the modelled concentrations are close to zero in the presence of 2-butanol. For example, C731CO2H, KLIMONIC and KLIMONONIC are formed by ozone attack on the limona ketone which, in the model, is formed by the initial OH attack on the exocyclic double bond of limonene. Owing to the presence of 2-butanol, this attack on the double bond is reduced thereby minimizing the amount of products generated. The correlation results for the humid and dry cases show that C823CO3H, C823OOH and C825OOH are negatively correlated with OH levels in the model. The reaction with OH represents the only destruction pathway of the produced acids in the model (even if unsaturated). This negative correlation indicates that, as the OH levels increase, the OH-induced destruction of the respective acid dominates over acid production. However, the reactions of unsaturated acids with ozone are not included in the MCM.

The experimental results reveal that the four dominant compounds are $C_8H_{12}O_4$, $C_8H_{12}O_5$, $C_9H_{14}O_4$ and $C_9H_{14}O_5$. However, $C_9H_{14}O_3$ which plays only a minor role in the experiments, represents the dominant compound in the modelling results. $C_8H_{12}O_4$ which contributes significantly to the experimental results, is characterised by medium-level contribution to the model. $C_8H_{12}O_4$ and $C_8H_{12}O_5$ exhibit a positive OH-dependence in the model consistent with the gas-phase results obtained for $C_8H_{12}O_4$ under humid low-ozone and all medium-ozone experiments. The estimated concentration of $C_8H_{12}O_5$ is lower in the presence of OH chemistry for most conditions except for humid low ozone experiments. The model reveals a positive OH dependence and a negative OH dependence for $C_9H_{14}O_4$ and $C_9H_{14}O_5$, respectively. The behaviour of the $C_9H_{14}O_4$ gas phase concurs with the model results for low- and medium-ozone experiments. For the highest ozone-level experiments, the levels observed for mixed oxidation are lower than those observed for oxidation in the presence of an OH scavenger. The OH dependence of $C_9H_{14}O_5$ in the experimental results differs from the overall negative OH dependence of modelled $C_9H_{14}O_5$. In contrast to the model predictions, the C7 acids $C_7H_{10}O_4$ and $C_7H_{12}O_3$ contribute significantly to the gas-phase results and exhibit only a weak OH dependency. The model predicts a weak OH dependence for $C_{10}H_{16}O_3$ which is in stark contrast to the strong dependence revealed by the experimental results. Overall, most acids exhibit a positive RH dependence in the medium-ozone and humid low-ozone cases, a behaviour that is lacking from the modelling results. However, consistent with the modelling results, water in the system increases the concentration of $C_{10}H_{16}O_3$ by a factor of two. This hold true for all cases, except for the highest ozone cases where the concentrations observed in the experiments are higher than the values predicted for dry conditions. In conclusion, significant concentrations

of 11 acids (from a total of 16) included in MCM are measured with the CIMS. The model predictions are, in some cases, inconsistent with the measurement results, with the most notable inconsistencies occurring for the OH dependence.

## 4 Mechanism interpretation and outlook

The formation and the dependence of the eight most prominent ions in the experiments are only partly explained by the model and reaction pathways that form compounds with the molecular formulas $C_7H_{10}O_3$ and $C_{10}H_{16}O_4$ are absent. In the following, we propose reaction pathways for explaining the formation of some ions not accounted for in the model and propose additional pathways for compounds already included in MCM. Examples include reactions of unsaturated acid products with ozone or the formation of C10 acyl radicals via the hot acid channel (see pathway A in Figure 1). The largest discrepancy between model and experimental results is observed for the formation of compounds, such as the group of C7 acids or ketolimononic, -limonic or -limonalic acid which are OH-dependent in the model but are OH-independent in the observations. The dominance of $C_9H_{14}O_4$ and positive correlation with ozone can be explained by the additional formation of ketolimononic acid via reaction pathways as seen in Figure . Here, ozone attacks the double bond of the primary product limononic acid thereby forming a CI. In the case of exocyclic CI, sCI can be formed directly and the remaining CI* are usually more effectively stabilised than endocyclic CI* and therefore a larger yield of sCI can be expected. The produced sCI can produce ketolimononic acid via the water channel (seefigure 5). Ketolimonalic ($C_8H_{12}O_4$) and ketolimonic ($C_8H_{12}O_5$) acid may be formed via the reaction of limonalic (R2, $C_9H_{14}O_3$) and limonic acid (R3, $C_9H_{14}O_4$), respectively, with ozone. The formation of a vinyl hydro peroxide (VHP) and subsequent decomposition via OH elimination and oxygen addition to the alkyl radical yields an alkyl peroxy radical. The bimolecular reaction of the alkyl peroxy with other $RO_2$ can lead to an alkoxy radical which then can form a carbonyl and $HO_2$ upon reacting with oxygen. This reaction chain may explain the formation of $C_9H_{12}O_5$ and $C_8H_{10}O_{5-6}$.

The model predictions for cases with and without the scavenger differ only slightly but the reaction pathway involving OH is an important contributor to $C_{10}H_{16}O_3$ formation in the experiments. This becomes especially clear when dry experiments with/without OH (with no possibility for the water pathway) are compared. The pathway leading to the formation of $C_{10}H_{16}O_3$ via the hot acid channel from the anti – CI* (see Figure 1) is also neglected by the model. Figure 6 illustrate how the remaining double bond can also be attacked by OH which would lead to the formation of an alkyl radical and subsequent addition of $O_2$. The reaction pathways shown in figure 6 lead to the observed acid formation and may explain the formation of $C_7H_{10}O_4$, $C_9H_{14}O_5$ and $C_9H_{14}O_4$. The produced

alkoxy radical will probably follow pathway A which produces the most stable radical and subsequently $C_7H_{10}O_4$. This pathway involves two bimolecular steps and is positively correlated with $RO_2$ levels in the system. Saturated compounds, although non-reactive with ozone, are susceptible to secondary chemical reactions induced by OH. Figure 6ii illustrate how the fate of the saturated compounds will depend on the relative reactivity of different sites to OH, and may include the abstraction of the acidic hydrogen followed by splitting off of $CO_2$; the subsequent bimolecular reactions will produce $C_7H_{10}O_4$. This reaction competes with the abstraction of the tertiary hydrogen, but will lead to products that are inconsequential to the present experimental results. The formation of $C_{10}H_{16}O_4$ results from processes other than ozonolysis or OH attack on the exocyclic double bond of an acid product due to the fragmentation of the produced POZ and excessively high resulting O numbers. $C_{10}H_{16}O_4$ may have resulted from the reaction of an acylperoxy radical with $HO_2$ (see Figure 1) and the formation of a peroxy acid. However, the pathway for $C_7H_{10}O_3$ formation remains unclear. For the particle phase, $C_8H_{12}O_5$ and $C_9H_{14}O_5$ are the dominating compounds in most experiments performed in this study, whereas $C_{10}H_{16}O_3$ (a major gas-phase compound) represents only a minor contributor to this phase. $C_{10}H_{16}O_4$ is excluded in the model, but plays a role in the particle-phase results. The formation of $C_{10}H_{16}O_4$ is positively correlated with the presence of $RO_2$ and $HO_2$. $C_9H_{14}O_3$ and $C_9H_{14}O_4$ are the dominant acids in the model calculations, but are only minor compounds in the particle phase. $C_9H_{14}O_4$ formation seems to occur only in experiments with the highest limonene content. Rapid autoxidation for the formation of highly oxidised molecules (HOMs) has recently gained significant attention (Ehn et al., 2014). This autoxidation proceeds via intramolecular H abstraction of $RO_2$ and subsequent formation of hydroperoxide groups. $RO_2$ lifetimes in low NOx environments are usually sufficient for the occurrence of this process (Orlando and Tyndall, 2012). During this process, large amounts of oxygen are rapidly introduced into the molecules, leading to a decrease in their vapour pressure. Most of the $RO_2$ will originate from the VHP channel, in the case of limonene ozonolysis, and products will probably be non-identifiable unless the radical termination reaction yields a carboxylic acid. Even if the formed compounds contain one or more carboxylic acid groups, the corresponding low vapour pressure may be undetectable by the used FIGAERO inlet. Jokinen et al. (2014) investigated the formation of HOM from limonene and found that highly oxygenated monomers (C10) and dimers (C20) with oxygen numbers ranging from 5 to 11 and 7 to 18, respectively, play a crucial role in this formation. Only one compound with the same chemical formula ($C_9H_{14}O_5$) has been found in this study but it is unclear if the chemical structure is the same.

In this study, dimers have exclusively been detected in the particle phase and are absent from the gas phase, owing to their potentially low vapour pressure. The formation of dimer esters from $\alpha$-pinene ozonolysis has recently been investigated (Kristensen et al., 2016). In that work, the reaction of sCI with carboxylic acids, suggested as the

formation pathway in the gas phase, was followed by partitioning into the particle phase. Consequently, the carboxylic acid group is lost in the esterification process which may explain the relatively low signals observed for acidic dimers in the present study. A potential acidic dimer ester will only be detectable if the dimer has a carboxylic acid group, as in the case of a di or tri carboxylic acid, or if the sCI carries a carboxylic acid group. Unsaturated dimers may react with ozone. The $C_{18}H_{28}O_8$ and $C_{19}H_{30}O_7$ can form via the reaction of the endocyclic limonene sCI with ketolimonic ($C_8H_{12}O_5$) or limonic ($C_9H_{14}O_4$) acid, respectively (see Fig. 7). In addition, $C_{19}H_{30}O_8$ may be formed from the dimerization reaction of limononic acid and the limononic-sCI. Gas-phase dimerization reactions of dominating C7–C10 acids with sCI account for only some of the dimer formulas. Reactions of acids with relatively small carbon numbers (<C7), $RO_2$ dimerization reactions or condensed-phase reactions may account for the other formulas.

## 5 Conclusion

Figure 8A provides an overview of the most important acidic compounds found in this study. These are identified by comparing the average contribution of each compound to all 33 experiments. Explicit formation pathways for the compounds $C_{10}H_{16}O_4$ and $C_9H_{12}O_5$ implemented in the model and additional reaction pathways for $C_7H_{10}O_4$, $C_8H_{12}O_{4-5}$, $C_9H_{14}O_{4-5}$ and $C_{10}H_{16}O_3$ are proposed. Structures for $C_{10}H_{14}O_5$ and $C_{10}H_{16}O_4$ have been proposed in previous studies (Jaoui et al., 2006;Rossignol et al., 2012;Leungsakul et al., 2005a;Walser et al., 2008;Marianne Glasius et al., 2000) but the current mechanistic understanding is inadequate for explaining the formation of compounds with the proposed structures. In Fig. 8A, we show that the mechanisms proposed in this work can improve the qualitative understanding of the formation characterising (on average) 65% of the dominant gas-phase compounds and 50% of the particle-phase compounds. Notably, the particle-phase data correspond partly to compounds with low oxygen content (2–3 Oxygen) and their formation and negative correlation with ozone remain unclear and require further study. However, only a few acidic dimers are detected. This may have resulted from lack of evaporation of these dimers (i.e. as acidic dimers) or loss of the acid functional group from potential acid monomer precursors during the dimer formation, as suggested in previous studies (Kristensen et al., 2016;Witkowski and Gierczak, 2014;Wang et al., 2016). Experiment #1, performed at low concentrations, for mixed oxidants and under humid conditions, should best represent atmospheric conditions. The summarised signal of the highest 10 acids in experiment #1 can be qualitatively attributed to 89% of the gas phase and the proposed mechanisms in this study account for 74% of the total signal (see Fig. 8B). The particle-phase composition can be qualitatively explained (by up to 42%) by the mechanisms proposed in this work. The relatively large percentage

of unexplained signal in the atmospheric case will result in large uncertainties when the acidic-particle phase composition of limonene SOA is modelled based on existing mechanisms (e.g. MCM) and partitioning theory. For a more quantitative mechanism (compared with the mechanism considered), inclusion of non-acidic products is required for a complete picture of the oxidation products. Furthermore, secondary and tertiary chemistry must be considered when the oxidation of compounds is modelled. Subsequent aerosol formation as well as dimerization and condensed-phase reactions must also be evaluated.

**Table 1:** Summary of experimental conditions and overview of the selected results. OH, $HO_2$ and $RO_2$ are integrated concentrations (ppb×s) as calculated with the model using the MCM. Water in the system increases the SOA mass and yield in the system whereas the presence of 2-butanol decreases the SOA mass and yield. The contribution of the highest 10 compounds is linked to complexity of the product distribution, i.e. if 10 compounds dominate then any minor product is less important and the composition appears less complex. The rate normalised $HO_2/RO_2$ ratio assumed rate coefficients of 8.8 x $10^{-13}$ and 9.1 x $10^{-12}$ for $RO_2$ and $HO_2$ reaction respectively (MCM v3.3.1)

| # | $O_3$ (ppb) | limonene initial [$\Delta$reacted] (ppb) | RH | OH – S | OH (ppb×s) | $HO_2$ (ppb×s) | $RO_2$ (ppb×s) | Rate normalised* [$HO_2$]/[$RO_2$] | SOA Mass ($\mu g\ m^{-3}$) | Yield of $c_7$-$c_{10}$ carboxylic acids | % Contribution of highest 10 ions in gas [particle] phase |
|---|---|---|---|---|---|---|---|---|---|---|---|
| 1 | 400 | 15 [ 8,4] | 40% | - | 0.085 | 1 | 400 | 0.03 | 2.6 | 23% | 73 [54] |
| 2 | 400 | 40 [22] | 40% | - | 0.083 | 3 | 710 | 0.04 | 4.2 | 18% | 73 [57] |
| 3 | 400 | 150 [81] | 40% | - | 0.079 | 5 | 1500 | 0.03 | 32 | 20% | 72 [51] |
| 4 | 400 | 15 [8,4] | Dry | - | 0.086 | 2 | 400 | 0.05 | 0.04 | 11% | 79[a] |
| 5 | 400 | 40 [22] | Dry | - | 0.084 | 3 | 710 | 0.04 | 0.8 | 4% | 80[a] |
| 6 | 400 | 150 [81] | Dry | - | 0.079 | 5 | 1500 | 0.03 | 12 | 2% | 78[a] |
| 7 | 400 | 15 [5,7] | 40% | ✓ | 0 | 10 | 190 | 0.55 | 2.3 | 13% | 69 [61] |
| 8 | 400 | 40 [15] | 40% | ✓ | 0 | 16 | 330 | 0.50 | 6 | 7% | 72 [60] |
| 9 | 400 | 150 [54] | 40% | ✓ | 0 | 29 | 640 | 0.47 | 40 | 6% | 70 [58] |
| 10 | 400 | 15 [5,7] | Dry | ✓ | 0 | 10 | 190 | 0.55 | 0.2 | [a] | 91[a] |
| 11 | 400 | 40 [15] | Dry | ✓ | 0 | 19 | 330 | 0.60 | 2.3 | [a] | 90[a] |
| 12 | 400 | 150 [54] | Dry | ✓ | 0 | 29 | 640 | 0.47 | 57 | [a] | 87[a] |
| 13 | 1000 | 15 [13] | 40% | - | 0.160 | 2 | 530 | 0.04 | 4.7 | 18% | 74 [53] |
| 14 | 1000 | 40 [34] | 40% | - | 0.157 | 4 | 930 | 0.04 | 7.3 | 9% | 75 [54] |
| 15 | 1000 | 150 [124] | 40% | - | 0.152 | 7 | 1900 | 0.04 | 25 | 7% | 71 [49] |
| 16 | 1000 | 15 [13] | Dry | - | 0.162 | 2 | 530 | 0.04 | 0.04 | 8% | 75 [63] |
| 17 | 1000 | 40 [34] | Dry | - | 0.160 | 4 | 9200 | 0.00 | 1.4 | 4% | 77 [51] |
| 18 | 1000 | 150 [125] | Dry | - | 0.154 | 7 | 1800 | 0.04 | 19 | 7% | 73 [51] |
| 19 | 1000 | 15 [10] | 40% | ✓ | 0 | 13 | 280 | 0.48 | 13 | 18% | 63 [51] |
| 20 | 1000 | 40 [28] | 40% | ✓ | 0 | 20 | 460 | 0.45 | 18 | 10% | 64 [55] |
| 21 | 1000 | 150 [101] | 40% | ✓ | 0 | 38 | 890 | 0.44 | 94 | 8% | 61 [55] |
| 22 | 1000 | 15 [10] | Dry | ✓ | 0 | 13 | 280 | 0.48 | 0.9 | 6% | 70 [61] |
| 23 | 1000 | 40 [28] | Dry | ✓ | 0 | 21 | 460 | 0.47 | 6.3 | 4% | 72 [58] |

| 24 | 1000 | 150 [101] | Dry | ✓ | 0 | 33 | 890 | 0.39 | 66 | 6% | 69 [55] |
|---|---|---|---|---|---|---|---|---|---|---|---|
| 25 | 5000 | 15 [15] | 40% | - | 0.242 | 3 | 570 | 0.05 | 4 | 8% | 60 [52] |
| 26 | 5000 | 40 [40] | 40% | - | 0.254 | 4 | 1000 | 0.04 | 10 | 4% | 60 [47] |
| 27 | 5000 | 150 [150] | 40% | - | 0.253 | 8 | 2000 | 0.04 | 56 | 2% | 56 [49] |
| 28 | 5000 | 15 [15] | Dry | - | 0.248 | 3 | 550 | 0.06 | 4 | [a] | 78[a] |
| 29 | 5000 | 40 [40] | Dry | - | 0.260 | 4 | 980 | 0.04 | 5.7 | [a] | 78[a] |
| 30 | 5000 | 150 [150] | Dry | - | 0.259 | 8 | 1900 | 0.04 | 22 | [a] | 75[a] |
| 31 | 5000 | 15 [15] | 40% | ✓ | 0 | 12 | 340 | 0.37 | 23 | 23% | 62 [51] |
| 32 | 5000 | 40 [40] | 40% | ✓ | 0 | 19 | 530 | 0.37 | 35 | 10% | 65 [52] |
| 33 | 5000 | 150 [150] | 40% | ✓ | 0 | 37 | 1000 | 0.38 | 186 | 7% | 64 [58] |

[a] Some data for experiments 4-6, 10–12 and 28–29 are unavailable due to no/low particle concentrations or malfunctioning of the FIGAERO unit.

**Figure 1.** (A) Example of initial reactions of limonene with ozone to form limononic acid from the anti – CI* via the hot acid channel and (B) the collisional stabilisation channel (Vereecken and Francisco, 2012) .

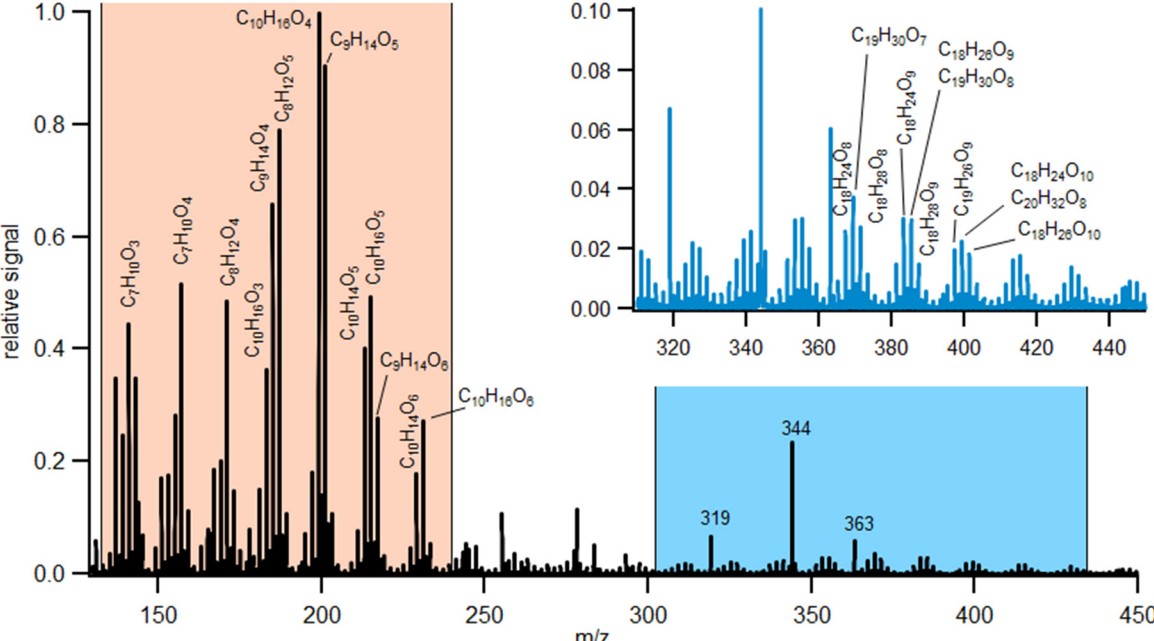

**Figure 2.** Example of derived mass spectra of condensed phase taken from the experiment (#21) with medium-ozone and high-limonene conditions with added OH scavenger. Indicated are the regions with identified monomers (orange region) and selected dimers (blue region). The peaks at 319, 344 and 363 are associated with the used mass calibrant HPFA. The complexity of this mass spectra is described using the fraction of 10 dominated product ions and are compared to other experiments with similar conditions in Fig. 3.

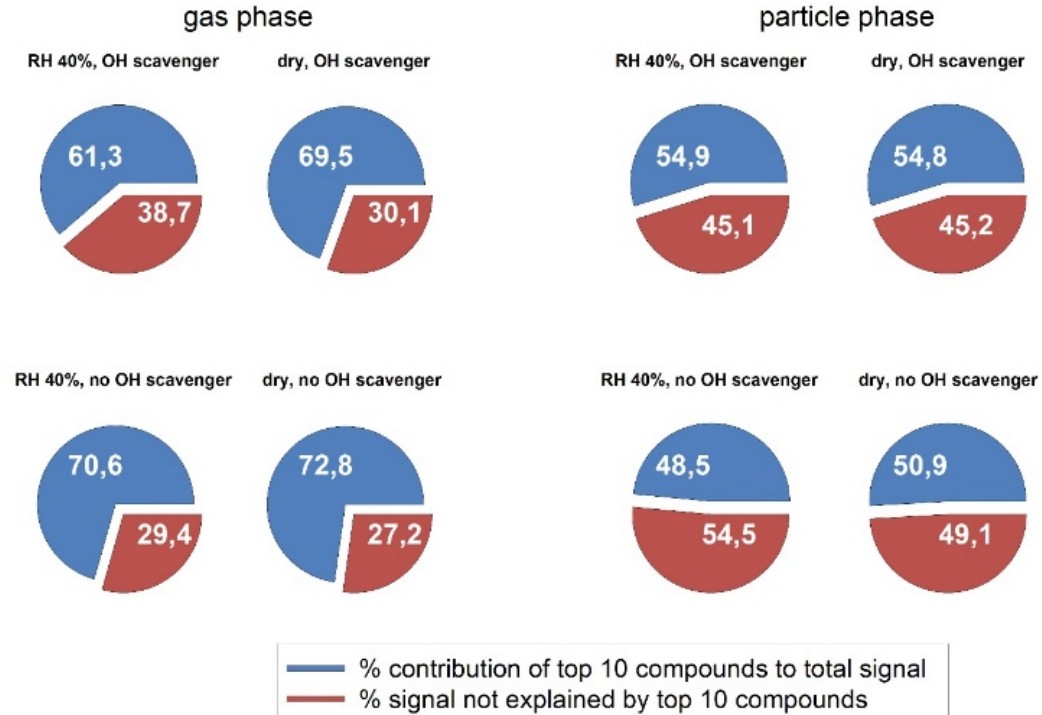

**Figure 3.** Contribution of the highest 10 compounds (blue) to the total signal in the gas and particle phase. The red fraction shows the signal not explained by the 10 highest compounds. A larger red fraction would indicate a more complex composition. Data shown for selected experiments with 1000 ppb ozone and 150 ppb limonene under different conditions. The estimated total yield of larger carboxylic acids (c7-c10) for these four experiments were very similar 6% (dry, scavenger), 8% (humid, scavenger), 6% (dry, no scavenger) and 7% % (humid, no scavenger).

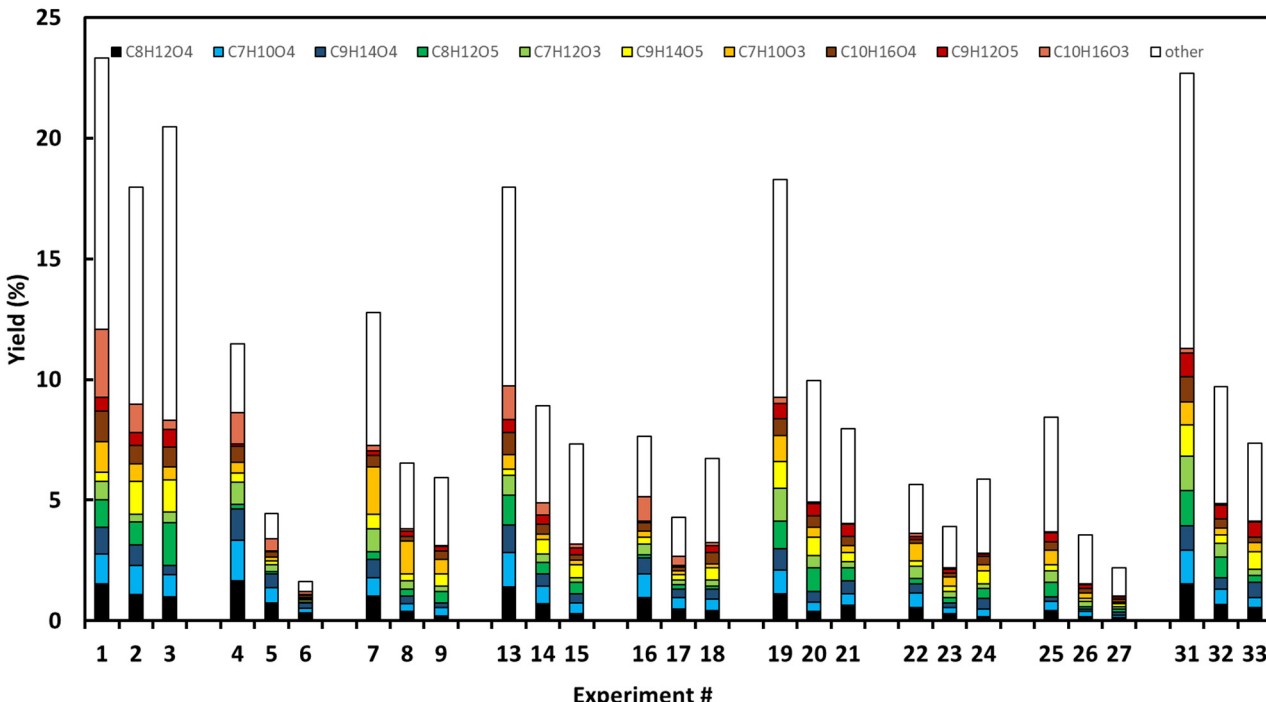

**Figure 4.** The molar yields of identified c7-c10 acids. The most important acids (averaged over all experiments) are $C_7H_{10}O_3$, $C_7H_{10}O_4$, $C_8H_{12}O_4$, $C_8H_{12}O_5$, $C_9H_{14}O_4$, $C_9H_{14}O_5$, $C_{10}H_{16}O_3$ and $C_{10}H_{16}O_4$ and are illustrated by individual colours. Exp. 10-12 and 28-30 were removed due to uncompleted particle phase characterisation.

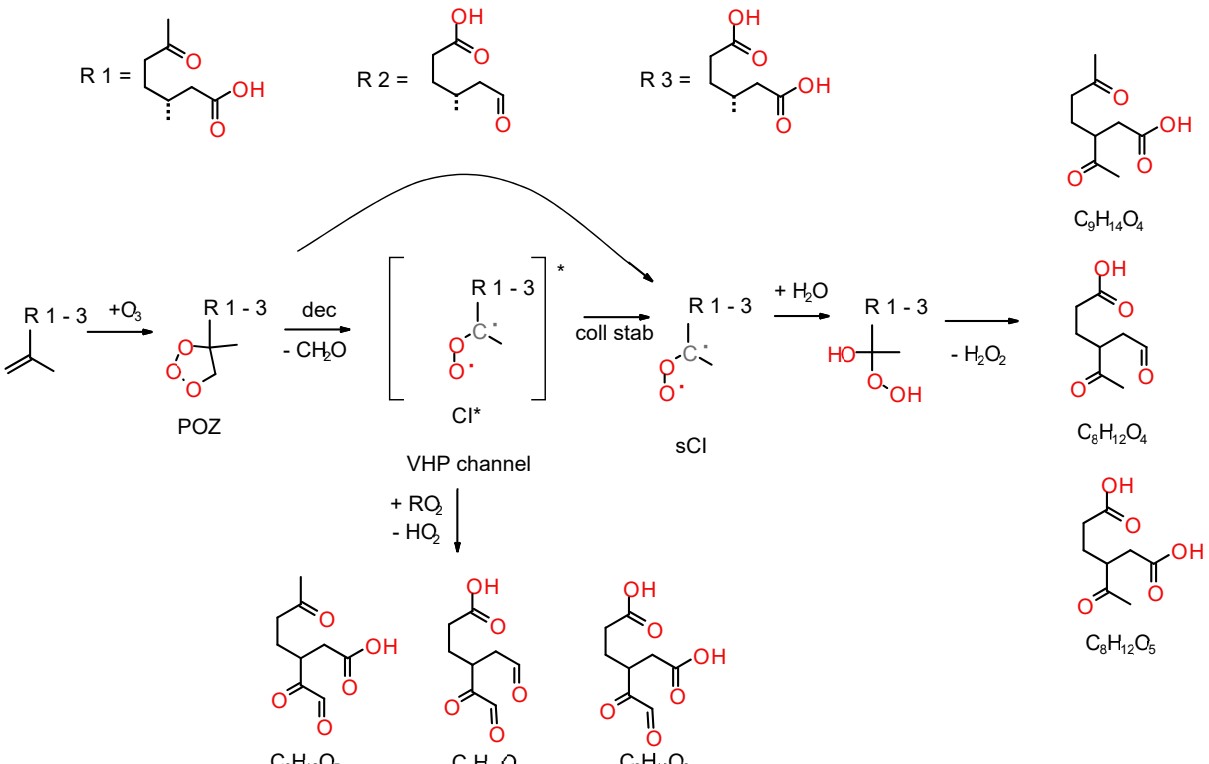

**Figure 5.** Proposed reaction mechanisms for secondary ozone chemistry of limononic (R1), limonic (R2) and limonalic (R3) acid. The dotted bond shows where R1-3 is connected to the 2-methyl-etenyl group (i.e. –C(CH$_3$)=CH$_2$) in the parent compounds.

**Figure 6.** Secondary chemistry i) Addition of OH to the unsaturated double bond of the primary product limononic acid and the subsequent formation of $C_7H_{10}O_4$, $C_9H_{14}O_5$ and $C_9H_{14}O_4$. ii) Secondary chemistry of a selected saturated carboxylic acid product ($C_8H_{12}O_5$) giving observed $C_7$ products

**Figure 4.** Proposed formation of observed dimers ($C_{18}H_{28}O_8$ , $C_{19}H_{20}O_7$ and $C_{19}H_{30}O_8$) from monomer-crigee reactions.

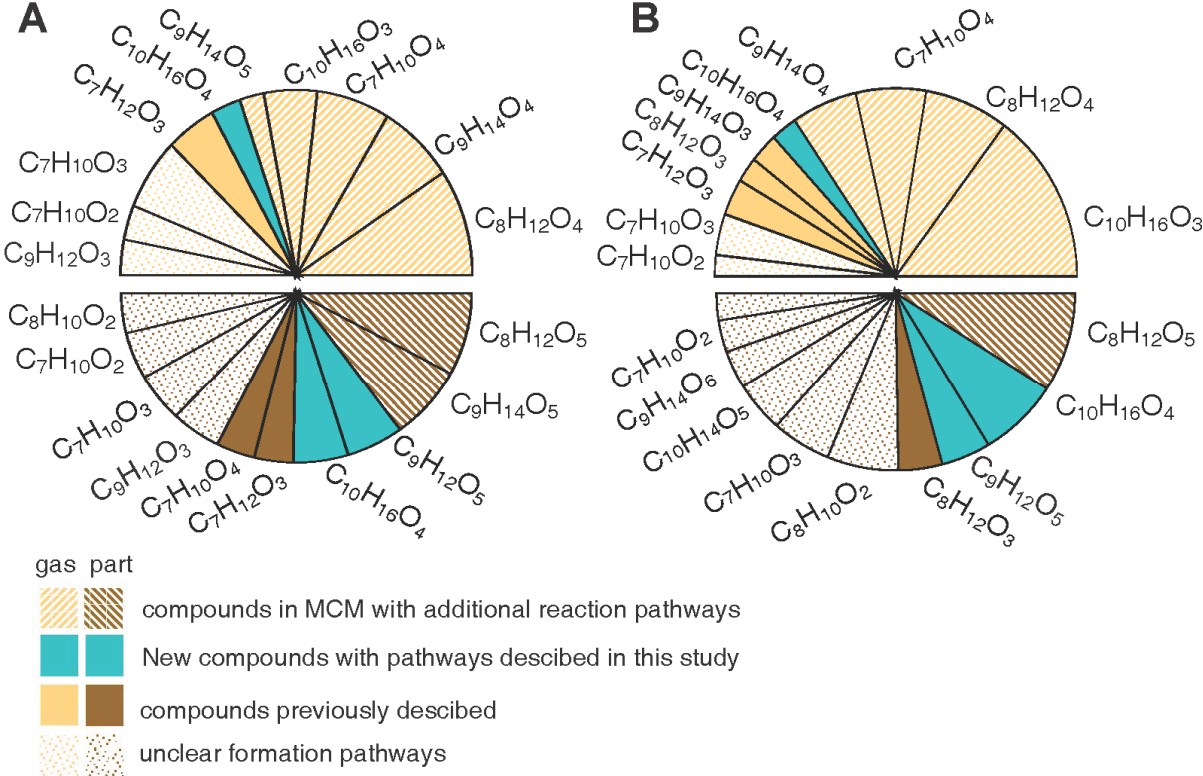

**Figure 8.** Pie charts showing the percentage contribution of each compound for the major 10 compounds observed. Top half wheel shows the gas-phase data and the bottom wheel shows the particle-phase data. The compounds are classified according to current knowledge, i.e. previous described reaction mechanism, mechanism suggested in this study and unclear formation pathways. A) Averaged contribution for all experiments. B) Specific contributions for experiment 1 that had the lowest concentrations of reactants, with mixed oxidants and humid conditions.

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
