# Peer review of "Carboxylic acids from limonene oxidation by ozone and OH radicals: Insights into mechanisms derived using a FIGAERO-CIMS"

_Atmospheric Chemistry and Physics, 2018_

## Referee Comment (RC1) · Anonymous Referee #1 · 25 Nov 2018

**General comments**

The manuscript presents the study of carboxylic acid formation from limonene ozonolysis. Experiments have been performed in a laminar- flow reactor in the dark under NOx-free conditions at 20°C, using various conditions of humidity, initial ozone and precursor concentrations, and with or without the use of an OH scavenger. The gas and particle phases were analyzed using an acetate HR-ToF-CIMS for the measurement of carboxylic acids. Around 100 molecular formulas of carboxylic acid have been identified, the chemical structures have been suggested for the major detected carboxylic acids, and their contribution to the total carboxylic acid signal has been calculated. Spearman correlation analysis and comparisons with the MCM have been performed.

[Figure]

Reaction pathways have been suggested to explain the formation of some carboxylic acids no present in the MCM. The work performed here provides a large and original experimental dataset on carboxylic acid formation from limonene ozonolysis. From my point of view, the manuscript still need large improvements to provide a clear message and an argued discussion. The following points have to be considered before publication.

**Major comments**

1. The discussions in section 3 and 4 of the manuscript should (1) be supported by the experimental/modeling work performed here showing appropriated figures, (2) presented in a quantitative way and (3) compared to recent bibliographic references (especially from other research teams). These two sections of the manuscript should according to me be rewritten in this way. If not, the discussions appear subjective. Here is only one example among others in the manuscript on the sensitivity of carboxylic acid formation to humidity, initial ozone and precursor concentrations, with or without the use of an OH scavenger. Currently, the authors discuss the sensitivity in term of signal intensity, diversity of products... but the discussion remains qualitative (increase or decrease, considerable or slight, higher or lower, opposite effects, explained or unexplained...) and is not directly supported by figure 3 (showing only in percent the total contribution of the 10 major carboxylic acids to the detected signal for dry and humid, and with and without the OH scavenger). A quantitative discussion, supported by a figure that summarizes all of the 33 experiments, showing the measured carboxylic acid signal intensity, and the individual contribution of the major carboxylic acid molecular formulas could be of large interest here. The authors could for example present a figure showing for each experiment, as cumulated bar plots, the total signal intensity of the detected carboxylic acids and the contribution of the individual top 10 (or 20, ...) to the total signal intensity (in intensity not in %) (or to the total signal intensity divided by the reacted precursor quantity, as yields, to be able to compare more easily the different experiments?) (a) for the gas phase and (b) for the particulate phase.

2. The authors state at several places in the manuscript that a large amount of carboxylic acids is formed during limonene ozonolysis but the contribution of the detected carboxylic acids is never compared to the total amount of secondary organic species formed during the experiments. Would it be possible to quantify this? This quantification is indeed difficult on a concentration basis but could maybe be done on a carbon basis, i.e. carbon concentration in the detected carboxylic acids divided by the carbon concentration in reacted limonene amount (considering that the intensity of the signal is directly proportional to the concentration with the same proportional factor used for all the acids if possible?).

3. Spearman correlation analysis have been performed to interpret the results. I am personally not familiar with this analysis. At the reading of the manuscript, I am not convinced by the relevance of such a statistical criterion for the purpose of this study (for an experimental work or a modeling study) nor by the substantial interest provided in the interpretation of the spearman correlations (the conclusions being mainly that two variables have a positive or negative correlation). Could the authors explain their objectives prior showing the spearman correlations? Have the spearman correlations been used previously for nonlinear / multigenerational / atmospheric chemistry? Also, I find these figures rather complex so could the authors discuss in general what we learn for a few selected points (what does it mean and what do we learn for example if the correlation is -1, 0 or 1?) Can we talk about a correlation between two variables if the spearman correlation is close to 0? If the results from these spearman correlations and rank correlations analysis is of largest interest for the manuscript, figures should be shown in the manuscript and not in supplementary, and they should be clearly presented and longer discussed.

4. For the comparisons performed between the MCM and the experiments, more information and justifications should be provided in the manuscript. In particular, could the authors explain how the model has been set to represent the experiments (box-model used, representation of the gas/particle partitioning, estimation of the vapor

pressures, initialization...)? Could a simulated temporal evolution be shown in the manuscript for a typical experiment? Also, I would have expected a comparison between model/measurement rather than a spearman correlation between MCM species. Could a figure summarizing quantitatively the MCM/experiment comparisons for the carboxylic acids be provided (for all experiments) in the manuscript and discussed in detail? One detail, the MCM is not a "model" as written several time in the manuscript but a chemical mechanism.

**Minor comments**

1. This paper focus on limonene ozonolysis and the experiments are performed in the dark under NOx-free conditions. I think this should be explicitly written somewhere in the manuscript.

2. To clarify the discussions (1) figures/tables should be presented and discussed once, before presenting the conclusions and comparisons to other studies and (2) the legend of the figures and the tables should be clearer / more precise. Here are a few examples only:

- p.5 l.5... "The general effect of parameters on SOA formation concurs with our previous results" but the results of this study have not been presented yet.

- p.5 l.16, p.5 l.19, p7 l.12... qualitative conclusions are provided with references to figures in parenthesis but figures have not been presented and discussed yet in the manuscript

- figure 4: please show which carbon of R1, R2 and R3 is connected to –C(CH3)=CH2

- the legend of Figure 8 is not clear (ex: compound previously described?)

- table S2: are the structure proposed by the authors or in literature?

- The legend of figure S3, S4 and especially S5 should provide more information

3. I think a discussion on the selectivity of the reagent ion (acetate) is needed somewhere. Are all the carboxylic acids detected and are all the detected species carboxylic acids, as suggested by the authors? Could some interferences occur with other species (such as organic peroxy acids formed in low-NOx conditions)? What is the possible impact of these interferences on the results of this study?

4. p.5 l.26 "the proposed structures of these acids are also shown" On which criteria are the structure proposed? Based on the "common" gaseous chemical pathways? On literature? A table with the carboxylic acid structures proposed by the authors should be included in the manuscript.

**Technical corrections**
p.1 l4: What "profile" are we talking about? Remove the word?

p.1 l.9: Should "The measured concentrations of dimers" be changed by "The measured concentration of dimers bearing at least one carboxylic acid function"?

p.1 l.15: I don't understand the meaning of this sentence (and not fully figure 8) "Based on the mechanisms proposed in this work, nearly 75% of the qualitative gas-phase signal of the low concentration (ppb converted), humid, mixed oxidant experiment can be explained"

p.2 l.4: Is this sentence used to justify why limonene is studied: "The emission rates of limonene are lower than those of other monoterpenes (e.g. a-pinene), and limonene is doubly unsaturated and exhibits high reactivity in the presence of ozone"?

p.2 l.7: Does "primary" in "However, the primary products may be unsaturated" mean emitted? Should "primary" be replaced by "first generation product"?

p.2 l.10: Please remove "basic" in "basic reaction mechanisms"

p.2 l.17: The sentence "The 10 carbon skeleton is retained during this process" is not right if O3 addition occurs on the exocyclic double bond

p.3 l.15: "The OH scavenger reduces the OH concentration but leads to an increase in

the HO2 concentration" and also in the RO2 concentration

p.4 l.14: Please refer to "table S1" after "A summary of experimental conditions is provided in"

p.4 l.16: A reference is needed here "The reagent ion acetate is especially susceptible to acidic organic compounds such as carboxylic acids"

p.4 l.19: Change "The gas-phase chemistry" into "The gas phase composition"?

p.5 l.6: Something is missing here "(for e.g.)"

p.5 l.31 to p.6 l.5: These sentences refer to the model and should be moved after the discussions on the experimental results

p.6 l.5: If the species are of low volatility they are not VOC (volatile organic compound)

p.6 l.9: Should "carbon number >= 10" be replaced with ""carbon number <= 10"?

p.6 l.13 Something is missing in "(e.g.)"

p.8 l.7 to p.8 l.13 The discussion on the HO2/RO2 ratio is not clear

p.9 l.23: Fig. 11 does not exist

p.17: Please, provide the  of the experiment and explicit "OH-S"

p.18: Does the figure show the "explained and unexplained fraction" (see title) or the "top 10 and other than top 10" (see labels)?

---

## Referee Comment (RC2) · Anonymous Referee #2 · 19 Dec 2018

General comments

The authors present results from 33 limonene oxidation experiments performed in a flow reactor at 20°C under different RH and precursor conditions. Using an acetate-FIGAERO-CIMs, they focus their analysis on carboxylic acids in the gas and particle phase. Molecular formulae and signal contributions of the most abundant acids are identified, also as a function of experimental conditions (humidity, OH scavenging, ozone level). Reaction mechanisms to explain inconsistencies between measurements and predictions based on the MCM are suggested.

The paper presents an interesting topic, and the dataset is very promising. However,

there are some issues with the analysis, and more importantly, the presentation of results, that need to be addressed before the paper can be considered for publication.

1. The motivation of the study could be carved out better. The authors introduce limonene as an important indoor VOC, but do not further go into detail about quantities etc. They then switch immediately to limonene in ambient air, but it is not clear at all whether indoor or outdoor limonene was the motivation for the study, and whether precursor conditions were tuned to simulate indoor or outdoor conditions (I assume, outdoor). In addition, the authors do not discuss any numbers, and only make qualitative statements in the introduction. This makes it hard to gauge the significance of the results presented here in the context of previous research.

2. A related point is to be made for the materials and methods section. A short discussion on precursor concentrations, and how they compare to atmospheric conditions, or other lab studies, would help placing the study in context within previous/ongoing research.

3. Generally, choices for both analysis and figures should be better motivated (scientifically). Why was this particular mass spectrum chosen in Figure 2? Why is the analysis focused on acids with 7 – 9 carbon atoms? Why were experiments with 1000 ppb of ozone and 150 ppb limonene only chosen for Figure 3? Again, the importance of the results presented here is hard to gauge without a clear scientific reasoning. In addition, e.g. Figure2 and Figure 3 would be more interesting if they represented average and diversity of mass spectra for different experiments, and not examples.

4. The discussion on the different effects on observed spectra is interesting, but somewhat hard to follow. It would be beneficial if the authors could provide figures that support their claims, and visualize the most important statements/relationships.

Specific comments

P. 1, l. 19: This reference is used rather often throughout the manuscript. It is without any doubt an important reference for the study. However, there might be certain subtopics of SOA that have seen some progress and update in the last decade, and it might be worth finding these.

P. 1, l. 19: Are number, size, and chemical composition "particle properties"? This sentence should be clarified.

P.2, l. 16: Why was only the anti – CI* pathway chosen for Figure 1, and not the syn – CI* pathway? Please motivate.

P. 2, l. 29: The switch to RO2 radicals is rather sudden. Try to better introduce that paragraph.

P. 4, l. 2: Table S1 should be moved into the main manuscript. It would greatly help in following the results.

P. 4, l. 17: Is it possible that dilution may have influenced the gas-particle equilibrium as resulting from G-FROST in your sampling line? From Figure S1 it looks like the flow to the SMPS was diluted as well (same inlet)?

P. 4, l. 32: Please motivate why you used Spearman correlation for your analysis.

P. 5, l. 16 – 17: Is the statement of the influence of water based on the numbers in Fig. 3? Are the differences significant?

P. 5, l. 23 – 24: Can the authors explain why the product distribution of the gas phase in pure ozonolysis experiments is more diverse?

Technical corrections

P. 1, l. 18 – p. 3, l. 32: The introduction should be divided into paragraphs, separated by line breaks.

P. 2, l. 16 – 17: "[...] where the oxygen atoms contribute to the formation of [. . .]"

P. 4, l. 4: Unfinished sentence

Caption Figure 2: Spell out OH-S

P. 9, l. 23: Should read "fig. 1", not "fig. 11"?

---

## Author Response (AR1)

Dear Editor,

We thank the reviewers for the helpful comments! We have now taken the benefit from those in improving the manuscript. A point by point response by a reply or/and actions (in black) to the reviewers' comments (in blue) will follow. New texts added or removed are shown in italics. The aim was to reply on each comment separately even if it creates some repetitions in replies (several comments solved by similar change). However, due to the number of comments there are a few that become obsolete due to changes from others. Then that are described by a reply. At the end is the full manuscript with changes in "track changes mode".

**Response to Anonymous Referee #1**

**General comments**

The manuscript presents the study of carboxylic acid formation from limonene ozonolysis. Experiments have been performed in a laminar- flow reactor in the dark under NOx-free conditions at 20◦C, using various conditions of humidity, initial ozone and pre- cursor concentrations, and with or without the use of an OH scavenger. The gas and particle phases were analyzed using an acetate HR-ToF-CIMS for the measurement of carboxylic acids. Around 100 molecular formulas of carboxylic acid have been identified, the chemical structures have been suggested for the major detected carboxylic acids, and their contribution to the total carboxylic acid signal has been calculated. Spearman correlation analysis and comparisons with the MCM have been performed.

Reaction pathways have been suggested to explain the formation of some carboxylic acids no present in the MCM. The work performed here provides a large and original experimental dataset on carboxylic acid formation from limonene ozonolysis. From my point of view, the manuscript still need large improvements to provide a clear message and an argued discussion. The following points have to be considered before publication.

**Major comments:**

1. 1.        The discussions in section 3 and 4 of the manuscript should (1) be supported by the experimental/modeling work performed here showing appropriated figures, (2) presented in a quantitative way and (3) compared to recent bibliographic references (especially from other research teams). These two sections of the manuscript should according to me be rewritten in this way. If not, the discussions appear subjective. Here is only one example among others in the manuscript on the sensitivity of carboxylic acid formation to humidity, initial ozone and precursor concentrations, with or without the use of an OH scavenger. Currently, the authors discuss the sensitivity in term of signal intensity, diversity of products... but the discussion remains qualitative (increase or decrease, considerable or slight, higher or lower, opposite effects, explained or un- explained...) and is not directly supported by figure 3 (showing only in percent the total contribution of the 10 major carboxylic acids to the detected signal for dry and humid, and with and without the OH scavenger). A quantitative discussion, supported by a figure that summarizes all of the 33 experiments, showing the measured carboxylic acid signal intensity, and the individual contribution of the major

carboxylic acid molecular formulas could be of large interest here. The authors could for example present a figure showing for each experiment, as cumulated bar plots, the total signal intensity of the detected carboxylic acids and the contribution of the individual top 10 (or 20, ...) to the total signal intensity (in intensity not in %) (or to the total signal intensity divided by the reacted precursor quantity, as yields, to be able to compare more easily the different experiments?) (a) for the gas phase and (b) for the particulate phase.

**Reply:** Thanks for the suggestions to clarify the presentation and discussion. The focus on relative intensities/concentrations is related to the aim to understand which acids are dominating under what conditions. Since there is always from laminar flow reactor studies hard to extract absolute yields we avoid explicit statements related to that and prefer to still a focus on relative intensities and correlations. Obviously, one can derive average yields/concentrations as measured at the end of the flow reactor, from the intensities presented in table s7 and s8 bearing in mind the residence time distribution.

**Action:** We have added a new Figure (now figure 4) where we present cumulated bar plots with the molar yield of the detected carboxylic acids (focus on the 8 most important acids with individually categories). We have added some quantitative information discussing some aspects of this figures and the overall molar yield from the experiments. Now these are presented in Table 1.

Text added:

"This experiment was done at medium ozone and high limonene concentration with an estimated 8 % molar yield of large carboxylic acids."

"Typically, these acids stand for 2-23% of reacted limonene on a molar basis assuming an average reaction time and CIMS sensitivity (Table 1). The distribution between gas and particle phase varied between compounds where the average particle fraction was between 5 and 80% depending on experimental conditions."

" Figure 3 shows for the comparable data to Fig. 2 (medium ozone, high limonene) the fraction of the 10 most prevalent ions. This was the most complete sub-step (dry-humid, with and without scavenger) of the total matrix and the illustrated pairwise features (e.g. dry vs humid) that was common also among other concentrations (e.g. for other condition we had missing particle phase data, see table 1). For these four selected experiments the estimated total yield of larger carboxylic acids (c7-c10) were very similar 6% (dry, scavenger), 8% (humid, scavenger), 6% (dry, no scavenger) and 7 % (humid, no scavenger)."

"Figure 4 shows the total molar yield of the acids identified in this study. One may note that the absolute yield presented here has significant uncertainties while the relative important of an acid can guide us in our mechanistic interpretation."

2. The authors state at several places in the manuscript that a large amount of carboxylic acids is formed during limonene ozonolysis but the contribution of the detected carboxylic acids is never compared to the total amount of secondary organic species formed during the experiments. Would it be possible to quantify this? This quantification is indeed difficult on a concentration basis but could maybe be done on a carbon basis, i.e. carbon concentration in the detected carboxylic acids divided by the carbon concentration in reacted limonene amount (considering that the intensity of the signal is directly proportional to the concentration with the same proportional factor used for all the acids if possible?).

**Reply:** Yes, using estimated limonene consumed, the intensities from Table S7 and S8 (now Table S2 and S3), and general sensitivity (from calibrations of standard acids) carbon yields can be estimated.

**Action:** To indicate the amount of c7-c10 acid produced the total molar yields estimated from experiments and modelling are now given.

*Beginning of abstract:

"The experiment and model providing yield of large (c7-c10) carboxylic acid in the order of 10% (2-23 and 10-15%, respectively)"

*Last sentence in abstract:

"Using the additional mechanisms proposed in this work nearly 75% of the observed gas-phase signal in our lowest concentration experiment (8.4 ppb converted, ca 23% acid yield) done at humid conditions can be understood."

*Discussion/result section a sentence was added:

"Generally, the model provides small variation in the total molar yield for the large carboxylic acids (c7-c10) of 10-15% while the experiments shows larger variability (2-23% molar yield) plausible due to the complication of aerosol formation not covered by the model."

3. Spearman correlation analysis have been performed to interpret the results. I am personally not familiar with this analysis. At the reading of the manuscript, I am not convinced by the relevance of such a statistical criterion for the purpose of this study (for an experimental work or a modeling study) nor by the substantial interest provided in the interpretation of the spearman correlations (the conclusions being mainly that two variables have a positive or negative correlation). Could the authors explain their objectives prior showing the spearman correlations? Have the spearman correlations been used previously for nonlinear / multigenerational / atmospheric chemistry? Also, I find these figures rather complex so could the authors discuss in general what we learn for a few selected points (what does it mean and what do we learn for example if the correlation is -1, 0 or 1?) Can we talk about a correlation between two variables if the spearman correlation is close to 0? If the results from these spearman correlations and rank correlations analysis is of largest interest for the manuscript,

figures should be shown in the manuscript and not in supplementary, and they should be clearly presented and longer discussed.

**Reply**: The Spearman correlation analysis was done as a tool to achieve a pattern on correlation that was used to extract selective compounds for more detailed analysis/description-maybe more as a guideline for our interpretation and as such we decided to give the full matrix. One could elaborate more on this but it will not be the main point of the paper.

**Action:**

*To reduce the focus on the spearman correlation we removed one sentence from the abstract:

**"**Spearman correlation analysis of the produced carboxylic acid species and experimental parameters were helpful in interpreting the results."

*To describe why we used spearman compared to normal correlation we added the following in the method section:

"A spearman correlation analysis was done based on of major products, experimental conditions and calculated radical concentrations. Compared to standard correlation the spearman correlation is more robust to outliers and independent of any assumptions about the distribution of the data. It was therefore preferred to assess the degree of association between each dominant acid and experimental parameters. The evaluation using spearman correlation is similar to other correlation methods giving 0, -1 and 1 for no correlation, perfect negative and positive correlation, respectively."

4. For the comparisons performed between the MCM and the experiments, more information and justifications should be provided in the manuscript. In particular, could the authors explain how the model has been set to represent the experiments (box- model used, representation of the gas/particle partitioning, estimation of the vapor pressures, initialization...)? Could a simulated temporal evolution be shown in the manuscript for a typical experiment? Also, I would have expected a comparison between model/measurement rather than a spearman correlation between MCM species. Could a figure summarizing quantitatively the MCM/experiment comparisons for the carboxylic acids be provided (for all experiments) in the manuscript and discussed in detail? One detail, the MCM is not a "model" as written several time in the manuscript but a chemical mechanism.

**Reply**: We use the existing chemical mechanism (MCM) to probe the effects on the formation of compounds due to changes in experimental conditions via gas-phase chemistry and the mechanism does not cover the particle phase. However, in comparison we use experimental data from both gas and particle phase. We do not show any temporal evolution since all measurements were done after the same residence time. However, one could use the model to illustrate formation of key compounds and e.g. radical distribution ($RO_2/HO_2$). Yes, the statement of MCM as a model is wrong and should be replaced by mechanism.

**Action:**

*We have added a sentence:
"The model did only consider gas-phase scheme of MCM."

*A section describing the model were moved from the first part of R&D to the model results and comparison section.

*We present the calculated integrated HO2/RO2 ratio (normalised to respective rate coefficient with RO2) in the experimental condition table (Table 1) as an example on variation in chemistry.
We have added another reference to this table to emphasis the modelling results.
 "However, the variation of radical distribution between experiments is illustrated in Table 1. Here the values of radicals are given as the integral concentration over the reaction time (unit pbb × s)."

'* We also added a discussion on Table 1- the text now reads:
"The model results show that the $HO_2/RO_2$ ratio in experiments employing the OH scavenger 2-butanol is approximately one order of magnitude higher than that of the mixed oxidant experiments. This higher ratio results from the $HO_2$ radicals generated by the reaction of 2-butanol with OH and will provide more influence of the $HO_2 + RO_2$ reaction in the experiment with scavenger. However, the $RO_2$ self-reaction are still the major pathway also in these experiment, twice the normalised rate of the $HO_2$ reaction. One may note that the $RO_2$ reaction rates are very much structural dependent and might be faster or slower that the assumed rates, see Jenkin et al. (2019) for a recent review on $RO_2$ chemistry."

*We added a comparison of the total acid yield:
"Generally, the model provides small variation in the total molar yield for the large carboxylic acids (c7-c10) of 10-15% while the experiments shows larger variability (2-23% molar yield) plausible due to the complication of aerosol formation not covered by the model."

*The MCM is now referred to as a mechanism rather than a model.

**Minor comments:**
1. This paper focus on limonene ozonolysis and the experiments are performed in the dark under NOx-free conditions. I think this should be explicitly written somewhere in the manuscript.
**Reply**:  We agree
**Action:** This is now included.
*The first sentence in the abstract:
"This work presents the results from a flow reactor study on the formation of carboxylic acids from limonene oxidation in the presence of ozone under NOx free conditions in the dark."

*The last sentence in introduction:

"This work (i) considers ozonolysis under dark condition and NOx free conditions (for various limonene concentrations) the effect of humidity, OH scavenging and ozone level on carboxylic acid formation,…"

2.To clarify the discussions (1) figures/tables should be presented and discussed once, before presenting the conclusions and comparisons to other studies and (2) the legend of the figures and the tables should be clearer / more precise. Here are a few examples only:
- p.5 l.5... "The general effect of parameters on SOA formation concurs with our previ- ous results" but the results of this study have not been presented yet.
- p.5 l.16, p.5 l.19, p7 l.12... qualitative conclusions are provided with references to figures in parenthesis but figures have not been presented and discussed yet in the manuscript
- figure 4: please show which carbon of R1, R2 and R3 is connected to – C(CH3)=CH2
- the legend of Figure 8 is not clear (ex: compound previously described?)
- table S2: are the structure proposed by the authors or in literature?
- The legend of figure S3, S4 and especially S5 should provide more information

**Reply**: 1) Yes, this is a valid point and should be implemented as far as possible. However, the two examples on this would rather reflect subjective ways of writing? Maybe we do not understand the points/examples correctly. The first point relates to a statement that the SOA behavior is in line with our previous study and we don't want to spend too much effort in repeating our old findings. Alternatively, we could remove this but then one would be criticised for not including any results on SOA formation behaviour. The second point is our way of writing a statement and then referring to a figure (in the same sentence) to support that statement. However, one could do it an opposite way saying "Figure x shows this and that". Maybe that would be clearer for the reader (as we interpret is the suggestion from the reviewer). 2). Yes, we should be more careful on doing this.

**Action:**
1) We removed the statements on SOA properties since that is not the focus on our paper. We have now rewritten part of the wording ensuring we introduce the figures more clearly.
2) The legend to the figures has been scrutinized and improved-especially for the points raised by the reviewer.

3. I think a discussion on the selectivity of the reagent ion (acetate) is needed somewhere. Are all the carboxylic acids detected and are all the detected species carboxylic acids, as suggested by the authors? Could some interferences occur with other species (such as organic peroxy acids formed in low-NOx conditions)? What is the possible impact of these interferences on the results of this study?

**Reply**: The acetate ionization is primarily sensitive for acids. However, the ionisation is also able to e.g. deprotonate nitrophenols as shown by Mohr et al. (2013); and Lopez-Hilfiker et al.

(2015) measured peroxy acids as their corresponding carboxylate anion in their study. In our study the detection of peroxy acids cannot be ruled out. So yes, the formation of peroxy acids would be an important interference and we do mention this as a possibility in the mechanism discussion but it should be more clearly written in e.g. materials and methods section.

**Action:** A statement on selectivity and potential interference from peroxy acids is given in the materials and methods section. We now use a sensitivity of $5.5 \times 10^{-3}$ Hz ppt$^{-1}$ (Le Breton et al, 2019) to calculate the total molar yield of the large carboxylic acids detected. This is now stated in the experimental section. Text read:

" One may note that also proxy acids has high sensitivities (Lopez-Hilfiker et al., 2015) and that the acetate ionisation has previously been used to detect nitro phenols (Mohr et al., 2013;Le Breton et al., 2019) and organic sulphates (Le Breton et al., 2019). However, here we assume carboxylic acids and peroxy acids to be the primarily compounds being observed with current set-up the CIMS. The used sensitivity for larger carboxylic acid was $5.5 \times 10^{-3}$ Hz ppt$^{-1}$ (Le Breton et al., 2019). This sensitivity was used to estimate molar yields; even if one should be precautious to provide absolute yields from this type of studies it provides indications on the product contribution."

4. p.5 l.26 "the proposed structures of these acids are also shown" On which criteria are the structure proposed? Based on the "common" gaseous chemical pathways? On literature? A table with the carboxylic acid structures proposed by the authors should be included in the manuscript.

**Reply**: The structures presented in the literature overview Table S1 were proposed by the authors for the paper referred to in that table.

**Action:** We more clearly direct the readers to Table S1:

"Table S1 also illustrate the proposed structures of these acids based on the current literature."

**Technical corrections:**
p.1 l4: What "profile" are we talking about? Remove the word?
**Action:** The word "profile" was removed

p.1 l.9: Should "The measured concentrations of dimers" be changed by "The measured concentration of dimers bearing at least one carboxylic acid function"?
**Action:** The beginning of the sentence was changed to

"The measured concentration of dimers bearing at least one carboxylic acid functional group".

p.1l.15: I don't understand the meaning of this sentence (and not fully figure 8) "Based on the mechanisms proposed in this work, nearly 75% of the qualitative gas-phase signal of the low concentration (ppb converted), humid, mixed oxidant experiment can be explained"
**Action:** The sentence was changed to:

"Using the additional mechanisms proposed in this work nearly 75% of the observed gas-phase signal in our lowest concentration experiment (8.4 ppb converted, ca 23% acid yield) done at humid conditions can be understood."

p.2 l.4: Is this sentence used to justify why limonene is studied: "The emission rates of limonene are lower than those of other monoterpenes (e.g. a-pinene), and limonene is doubly unsaturated and exhibits high reactivity in the presence of ozone"?

**Action:** The text now reads:

"..and elevated indoor concentrations can be expected (Brown et al., 1994;Langer et al., 2008) with subsequent SOA formation (Youssefi and Waring, 2014). The total global forest emission of limonene has been estimated to 11.4 Tg year$^{-1}$, placing it on the top four among monoterpenes (Guenther et al., 2012). A high aerosol yield and the two chemically different double bonds, an endocyclic and an exocyclic double bond makes limonene ozonolysis of specific interest (Koch et al., 2000;Saathoff et al., 2009;Chen and Hopke, 2010;Gong et al., 2018).."

p.2 l.7: Does "primary" in "However, the primary products may be unsaturated" mean emitted? "primary" be replaced by "first generation product"?

**Reply:** The word "primary" means first generation products in this context.

**Action:** The word "primary" was replaced by "first generation products"

p.2 l.10: Please remove "basic" in "basic reaction mechanisms"

**Action:** The word "basic" was removed

p.2l.17: The sentence "The 10 carbon skeleton is retained during this process" is not right if O3 addition occurs on the exocyclic double bond

**Reply:** The referee´s comment is correct, but in this case we refer to the attack on the endocyclic double bond.

**Action:** We clarified:

"The 10 carbon skeleton is retained during this process, if an endocyclic double bond is attacked."

p.3 l.15: "The OH scavenger reduces the OH concentration but leads to an increase in the HO2 concentration" and also in the RO2 concentration

**Action:** The sentence was changed to

"The OH scavenger reduces the OH concentration but leads to an increase in the $HO_2$ and $RO_2$ concentrations".

p.4 l.14: Please refer to "table S1" after "A summary of experimental conditions is provided in"

**Action:** The cross reference was added

p.4 l.16: A reference is needed here "The reagent ion acetate is especially susceptible to acidic organic compounds such as carboxylic acids"

**Action:** The reference to Bertram et al. 2011 was added

p.4 l.19: Change "The gas-phase chemistry" into "The gas phase composition"?

**Action:** "The gas-phase chemistry" was changed into "The gas phase composition"

p.5 l.6: Something is missing here "(for e.g.)"
**Action:** The sentence was removed based on another comment.

p.5 l.31 to p.6 l.5: These sentences refer to the model and should be moved after the discussions on the experimental results
**Action:** The sentences were moved

p.6 l.5: If the species are of low volatility they are not VOC (volatile organic compound)
**Action:** The text was changed accordingly: "..fraction of low-volatility and often very oxygenated organic compounds"

p.6 l.9: Should "carbon number >= 10" be replaced with ""carbon number <= 10"?
**Action:** The text was changed from "carbon number >= 10" to "carbon number <= 10"?

p.6 l.13 Something is missing in "(e.g.)"
**Action:** (e.g) was removed. "However, the relative signals are significantly lower than those reported for dimer formation in a study on limonene with nitrate radicals (Faxon et al., 2018) or the ozonolysis of other terpenes such as α-pinene (Kristensen et al., 2016)"

p.8 l.7 to p.8 l.13 The discussion on the HO2/RO2 ratio is not clear
**Action:** The discussion has been clarified and Table 1 now also includes a row of modelled HO2/ratios. the text now reads:

"The model results show that the $HO_2/RO_2$ ratio in experiments employing the OH scavenger 2-butanol is approximately one order of magnitude higher than that of the mixed oxidant experiments. This higher ratio results from the $HO_2$ radicals generated by the reaction of 2-butanol with OH and will provide more influence of the $HO_2$ + $RO_2$ reaction in the experiment with scavenger. However, the $RO_2$ self-reaction are still the major pathway also in these experiment, twice the normalised rate of the $HO_2$ reaction. One may note that the $RO_2$ reaction rates are very much structural dependent and might be faster or slower that the assumed rates, see Jenkin et al. (2019) for a recent review on $RO_2$ chemistry."

p.9 l.23: Fig. 11 does not exist
**Action:** The cross reference was changed to Fig. 1

p.17: Please, provide the of the experiment and explicit "OH-S"
**Reply:** The comment appears to be incomplete. We interpreted it as missing experimental number and writing clearly what OH-S stands for.
**Action:** Actions according to our interpretation but extended with more info in the caption:

"Example of derived mass spectra of condensed phase taken from the experiment (#21) with medium-ozone and high-limonene conditions with added OH scavenger. Indicated are the regions with identified monomers (orange region) and selected dimers (blue region). The peaks at 319, 344 and 363 are associated with the used mass calibrant HPFA. The complexity of this mass spectra is described using the fraction of 10 dominated product ions and are compared to other experiments with similar conditions in Fig. 3."

 Does the figure show the "explained and unexplained fraction" (see title) or the "top 10 and other than top 10" (see labels)?

**Reply:** The figure shows the contribution of the highest 10 compounds to the total signal.

**Action:** The caption was changed and are now reading:

"Contribution of the highest 10 compounds (blue) to the total signal in the gas and particle phase. The red fraction shows the signal not explained by the 10 highest compounds. A larger red fraction would indicate a more complex composition. Data shown for selected experiments with 1000 ppb ozone and 150 ppb limonene under different conditions. The estimated total yield of larger carboxylic acids (c7-c10) for these four experiments were very similar 6% (dry, scavenger), 8% (humid, scavenger), 6% (dry, no scavenger) and 7% % (humid, no scavenger)."

**Response to Anonymous Referee #2**

**General Comments:**

1. The motivation of the study could be carved out better. The authors introduce limonene as an important indoor VOC, but do not further go into detail about quantities etc. They then switch immediately to limonene in ambient air, but it is not clear at all whether indoor or outdoor limonene was the motivation for the study, and whether precursor conditions were tuned to simulate indoor or outdoor conditions (I assume, outdoor). In addition, the authors do not discuss any numbers, and only make qualitative statements in the introduction. This makes it hard to gauge the significance of the results presented here in the context of previous research.

**Reply**: Yes, one could elaborate more in the introduction. However, the general scientific motivation on studies of limonene ozonolysis cannot be divided by indoor or outdoor boundaries, but rather the difference in chemical regimes. Here e.g. the influence on RO2 chemistry could be important e.g. if the peroxy radical reacts with HO2 or via selfreaction (or in presence of NOx via NO).

**Action:**

*The introduction has been elaborated and now includes a quantitative motivation (we also added a point in method-see below). Text reads.

""..and elevated indoor concentrations can be expected (Brown et al., 1994;Langer et al., 2008) with subsequent SOA formation (Youssefi and Waring, 2014). The total global forest emission of limonene has been estimated to 11.4 Tg year$^{-1}$, placing it on the top four among monoterpenes (Guenther et al., 2012). A high aerosol yield and the two chemically different double bonds, an endocyclic and an exocyclic double bond makes limonene ozonolysis of specific interest (Koch et al., 2000;Saathoff et al., 2009;Chen and Hopke, 2010;Gong et al., 2018)."

*We have added the environmental condition (dark without NOx) in the abstract:

"This work presents the results from a flow reactor study on the formation of carboxylic acids from limonene oxidation in the presence of ozone under NOx free conditions in the dark."

In the last sentence of the introduction where we also state our aim on ambient atmospheric application:

"This work (i) considers ozonolysis under dark condition and NOx free conditions (for various limonene concentrations) the effect of humidity, OH scavenging and ozone level on carboxylic acid formation, and (ii) provides an outlook and suggestions for mechanistic gaps with the aim of eventually describing major acidic products found in the gas and particle phases under realistic atmospheric conditions, i.e. ozonolysis is performed in the absence of an OH scavenger under low concentration and humidity conditions."

2. A related point is to be made for the materials and methods section. A short discussion on precursor concentrations, and how they compare to atmospheric conditions, or other lab studies, would help placing the study in context within previous/ongoing research.

**Action:** Material and methods has been extended with a motivation on using this specific set-up and these conditions.

Text added:

"The experimental matrix was chosen to understand ozonolyis of limonene under dark NOx free conditions. The laminar flow reactor approach is well suited for investigating changes in experimental condition, e.g. dry vs humid, high vs low concentration. Due to short residence time the absolute concentration is higher than ambient even if the amount limonene converted (few pbb and higher) is similar to larger simulation chamber studies. The conversion of any $RO_2$ radicals is biased towards self-reaction; of importance in VOC dominated rural forest conditions and in the indoor environment."

3. Generally, choices for both analysis and figures should be better motivated (scientifically). Why was this particular mass spectrum chosen in Figure 2? Why is the analysis focused on acids with 7 – 9 carbon atoms? Why were experiments with 1000 ppb of ozone and 150 ppb limonene only chosen for Figure 3? Again, the importance of the results presented here is hard to gauge without a clear scientific reasoning. In addition, e.g. Figure2 and Figure 3 would be more interesting if they represented average and diversity of mass spectra for different experiments, and not examples.

**Action:** Motivations have been added to put the work/selected illustrations in context.

"Figure 3**Error! Reference source not found.** shows for the comparable data to Fig. 2 (medium ozone, high limonene) the fraction of the 10 most prevalent ions. This was the most complete sub-step (dry-humid, with and without scavenger) of the total matrix and the illustrated pairwise features (e.g. dry vs humid) that was common also among other concentrations (e.g. for other condition we had missing particle phase data, see table 1). For these four selected experiments the estimated total yield of larger carboxylic acids (c7-c10) were very similar 6% (dry, scavenger), 8% (humid, scavenger), 6% (dry, no scavenger) and 7 % (humid, no scavenger)."

4. The discussion on the different effects on observed spectra is interesting, but somewhat hard to follow. It would be beneficial if the authors could provide figures that support their claims, and visualize the most important statements/relationships.

**Reply**: Yes, the text could be complemented with a Figure.

**Action:** We have added a new Figure (now figure 4) where we present cumulated bar plots with the molar yield of the detected carboxylic acids (focus on the 8 most important acids with individually categories). We have added some quantitative information discussing some aspects of this figures and the overall molar yield from the experiments. Now these are presented in Table 1.

**Specific comments**

P. 1, l. 19: This reference is used rather often throughout the manuscript. It is with out any doubt an important reference for the study. However, there might be certain subtopics of SOA that have seen some progress and update in the last decade, and it might be worth finding these.

**Action:** We have complemented this with more recent citations.

P. 1, l. 19: Are number, size, and chemical composition "particle properties"? This sentence should be clarified.

**Action:** The sentence was clarified: "Atmospheric aerosol particles have an impact on climate and human health and the respective effects depend on particle properties determined by the size and its chemical composition"

P.2, l. 16: Why was only the anti – CI* pathway chosen for Figure 1, and not the syn – CI* pathway? Please motivate.

**Reply**: The anti – CI* pathway was chosen for Figure 1 because it shows the formation of carboxylic acids via the two pathways A and B. The syn-pathway would preferably decompose via the vinyl hydroperoxide channel

P. 2, l. 29: The switch to RO2 radicals is rather sudden. Try to better introduce that paragraph.

**Action:** A sentence was added to improve the transition:

"The decomposition of CI* can lead to the formation of alkyl radicals. They rapidly react with oxygen to form alkylperoxy radicals ($RO_2$) which are an important intermediate in the gas phase oxidation of organic compounds."

P. 4, l. 2: Table S1 should be moved into the main manuscript. It would greatly help in following the results.

**Reply**: We agree with the referee

**Action:** Table S1 was moved to the main manuscript (now Table 1)

P. 4, l. 17: Is it possible that dilution may have influenced the gas-particle equilibrium as resulting from G-FROST in your sampling line? From Figure S1 it looks like the flow to the SMPS was diluted as well (same inlet)?

**Reply**: Due to instrumental limitations dilution was necessary. An impact of dilution cannot be ruled out even if the residence time is rather short (<1 s). Most of the analysis was done by looking at sum of the compounds specified and not necessarily the partitioning.

**Action:** We have added a point regarding this potential interference for the most volatile compounds in the particulate phase.:

"The dilution was necessary for analytical reason and might evaporate some of the most volatile compounds from the condensed phase. However, the time between dilution and analysis was short (< 1s), exactly the same for all conditions and a possible effect would somewhat mimic atmospheric conditions."

P. 4, l. 32: Please motivate why you used Spearman correlation for your analysis.

**Reply**: Spearman correlation determines the strength and direction of the monotonic relationship between two variables. As a non-parametric rank correlation, Spearman analysis is independent of the assumption of any underlying distribution and was therefore chosen over Pearson correlation.

**Action**: To describe why we used spearman compared to normal correlation we added the following in the method section:

"A spearman correlation analysis was done based on of major products, experimental conditions and calculated radical concentrations. Compared to standard correlation the spearman correlation is more robust to outliers and independent of any assumptions about the distribution of the data. It was therefore preferred to assess the degree of association between each dominant acid and experimental parameters. The evaluation using spearman correlation is similar to other correlation methods giving 0, -1 and 1 for no correlation, perfect negative and positive correlation, respectively."

P. 5, l. 16 – 17: Is the statement of the influence of water based on the numbers in Fig. 3? Are the differences significant?

**Reply**: The statement is based on Fig. 3 as well as Table 1, which shows that the % contribution of the highest 10 compounds is lower in all humid experiment, which is interpreted as having a higher diversity

**Action**: We more clearly refer to Figure 3 and Table 1. The sentence now reads

"The presence of water in the system also increases the diversity of the product distribution in both the gas and particle phases, i.e. a small effect in Fig. 3 but consistent for all pairwise dry/humid experiments shown in Table 1."

P. 5, l. 23 – 24: Can the authors explain why the product distribution of the gas phase in pure ozonolysis experiments is more diverse?

**Reply**: This statement is an over simplification since there is a scatter in the diversity between different experiment. The sentence should be removed or more elaborated.

**Action:** The sentence was removed.

**Technical corrections**

P. 1, l. 18 – p. 3, l. 32: The introduction should be divided into paragraphs, separated by line breaks.

**Action:** The introduction was divided for better readability.

**Action:** The sentence was changed to

"The POZ will undergo decomposition where the oxygen atoms contribute to the formation of a carbonyl and a carbonyl oxide group, the so-called excited Criegee intermediate (CI*).

**Reply:** The sentence

[revised manuscript text omitted]